# Defect-evolved quadrupole higher-order topological nanolasers

Shengqun Guo ⓘ , Wendi Huang, Feng Tian, Yufei Zhou, Yilan Wang & Taojie Zhou ⓘ ✉

Topological photonics have been garnering widespread interest in engineering the flow of light with topological ideas. Strikingly, the recent introduction of higher-order topological insulators has generalized the fundamental framework of topological photonics, endowing counterintuitive strong confinement of light at lower-dimensional boundaries, thus unlocking exciting prospects for the exploration of topological phenomena in fresh routes as well as the design of topology-driven nanoscale light sources. Here, we revealed the photonic quadrupole topological phases can be activated by defect evolution and performed experimental demonstrations of associated nanoscale lasing operation under this paradigm. The quadrupole higher-order topological nanocavity is constructed by two topologically distinct photonic crystal slabs with opposite directions of defect evolution. Stable single-mode emission and low lasing threshold in telecom C-band are achieved at room temperature of the defect-evolved quadrupole topological nanolaser. This work reveals new possibilities for photonic quadrupole topological phase transition, providing an intriguing route toward light confinement and modulation under the topological framework.

The past few decades have witnessed the accelerated development of the fascinating area of topological insulators (TI). In general, the representative features of TI are the topological properties of the momentum space and the robust boundary states, which are reflected by the bulk-boundary correspondence[1–3]. Benefiting from the similarities between photonic crystal (PhC) and solid-state physics, analogous concepts extend into photonic systems, giving rise to the discipline of topological photonics[4–6]. Since the demonstration of quantum Hall effect edge states[7], topological photonics has emerged as a remarkable platform for exploring diverse exotic topological phenomena, such as Floquet photonic TIs[8], quantum spin Hall photonic TIs[9,10], valley Hall photonic TIs[11–13], and phase change material induced Chern topological phase transition[14]. In addition to the intrinsic importance for fundamental physics of topological aspect, topological photonics also provides unprecedented avenues for manipulating and confining light, thus opening a paradigm for the design of advanced photonic devices particularly in topological lasers, including but not limited to quantum Hall lasers[15,16], spin-momentum-locked edge state laser[17], topological valley hall laser[18], and the topological laser based on the bulk states of quantum spin Hall insulators[19]. Compared to the conventional laser architecture, topological lasers are prospected as a promising candidate for future on-chip high-performance coherent light sources with inherent robustness against imperfections and disorder[20].

Recently, a class of topological insulators, termed higher-order topological insulators (HOTI), has attracted significant attention due to its ability to host lower-dimensional boundary states (e.g., corner state) that go beyond the conventional bulk-boundary correspondence[21,22]. This line of research began with the prediction of multipole TIs[23,24], characterized by their quantized multipole moments. To obtain quadrupole TIs in two-dimensional (2D) systems that enable the emergence of highly localized 0D corner state, the critical approach is using the appropriate distribution of positive and negative couplings to introduce π-flux[23,24]. However, the practical implementation of this approach in photonic systems remains challenging and has so far been reported in limited platforms such as

School of Microelectronics, South China University of Technology, Guangzhou, China. ✉e-mail: taojiezhou@scut.edu.cn

coupled ring resonators[25] and waveguide arrays[26]. Consequently, significant research efforts have been devoted to alternative schemes in photonic systems for the quadrupole topology beyond the π-flux mechanism. Until yet, the potential for photonic quadrupole topological phases beyond the π-flux mechanism has not been fully explored, only a few approaches have been proposed, such as Floquet PhCs[27], magneto-optical PhCs[28,29], and twisted PhCs[30]. Additionally, the promising characterizations of topological corner state provide an innovative route to achieve lasing operation in an ultimate small scale with diffraction-limited mode volume ($V_m$) and ultra-low energy consumption. Nevertheless, the most extensively investigated higher-order topological corner state nanolasers are primarily based on Wannier-type HOTI with the expanded-shrunken scheme[31–35], which is a different class of symmetry-protected topological phases, where the Wannier centers are localized at maximal Wyckoff positions by manipulating the difference between inter- and intracell couplings[36,37]. The quadrupole topological nanolaser has only been demonstrated in a twisted PhC platform at a cryogenic temperature, mainly hindered by the implementation scheme of the photonics quadrupole topological phase[38]. Naturally, we wonder if there are new possibilities for realizing the photonic quadrupole topological phase and whether it can be applied to high-performance topological nanolasers, which is fundamentally crucial for exploring intriguing topological physics within the nanoscale platforms.

In this work, we propose a defect evolution method for realizing photonic quadrupole topological phases and experimentally demonstrate its topological features within 2D photonic crystal nanolasers. Specifically, the different branches of quadrupole topology are obtained by exploiting the evolution of the introduced geometrical defects in opposite directions (clockwise and anticlockwise). The corresponding 0D corner state is predictably located at the corner of two topologically distinguished semiconductor PhC slabs in quadrupole topology. Using InGaAsP multi-quantum wells as gain materials, we further experimentally demonstrate its stable single-mode lasing operation in quadrupole higher-order topological nanolasers in the telecommunication C-band. This work unveils a crucial method for realizing photonic quadrupole topological phase, establishing a new paradigm for the higher-order topological nanolaser design.

## Results

### Defect-evolved photonic quadrupole topological phase

Figure 1a shows a schematic diagram of the considered defect evolution for the square lattice PhC, where the original PhC structure without introducing defects is commonly employed to describe the 2D Su-Schrieffer-Heeger model with the topological feature of Wannier-type HOTIs[31]. Here, we consider air-hole type PhC thin slab for lasing purposes. In this scenario, the original PhC unit cell with lattice parameter $a$ consists of four square-shaped air holes with length $s = 0.3a$ located at ($\pm a/4$, $\pm a/4$). Afterward, to obtain the photonic quadrupole topological phases, the notch structural defects at the diagonal position for each original air hole are introduced, and the defect evolution in this context refers to the systematic geometric adjustments of the geometrical defects for air holes. The configuration describing the initial state of this process is illustrated in the left panel of Fig. 1a, where the side lengths $d_1$ and $d_2$ of introduced defects at the diagonal positions are correlated through a fixed total length $D = d_1 + d_2 = 0.5 s$ and equal to $d_1 = d_2 = D/2$ in the initial state. A defect-related parameter $\Delta d_1$ (in units of $s$) is used to describe the change of the length $d_1$ relative to the initial state in the evolved configuration. Unlike the manipulative strategy of breathing-type lattices[31,39,40], the topological properties of this photonic system we proposed here, are manipulated by the evolution of defects in different directions. When the commutative mirror symmetries $M_x$ and $M_y$ are removed by evolving the defect size either counterclockwise or clockwise, the

system maintains glide symmetries $G_x = (x, y) \rightarrow (-x + a/2, y + a/2)$ and $G_y = (x, y) \rightarrow (x + a/2, -y + a/2)$, the inversion symmetry $I = (x, y) \rightarrow (-x, -y)$, and the $C_4$ symmetry, providing the possibility of the quadrupole topological bandgap.

To explore quadrupole topological phases, we present the typical transverse-electric (TE) like band structure of PhC with representative clockwise evolution of defects at $\Delta d_1 = 0.25$ as shown in Fig. 1b. For the identical magnitude of $\Delta d_1$, the clockwise and counterclockwise configurations share same band structure due to the fact that they belong to mirror-symmetric partners. Nonetheless, the bandgap for unit cell with defects evolving in different directions possesses different topological properties. To elucidate this, Wannier band polarizations are utilized to describe their quadrupole topological phases. The $j$-th Wannier bands $v_y^j(k_x)$ can be obtained from the eigenvalues $e^{2\pi i v_y^j(k_x)}$ of Wilson-loop operator $W_{k,y} = \prod_{i=0}^{n_k-1} F_{k+i\Delta k_y}$[24,41], where the subscript $y$ indicates the considered path of the Wilson loop, and the matrix element for $F_k$ is $F_k^{nm} = \langle u_{n,k} | u_{m,k+\Delta k_y} \rangle$, with $u_{n,k}$ being the periodic part of the Bloch wavefunction. Further, the polarizations of Wannier bands can be determined by iteratively repeating the Wilson loop, i.e., nested Wilson loop along the orthogonal direction[24], where $u_{n,k}$ in Wilson loop is replaced by $w_{k,y}^j = \sum_{n=1}^{N_{occ}} u_{n,k} [\mathcal{E}_{k,y}^j]^n$, which is constructed by wavefunction and $n$-th element $[\mathcal{E}_{k,y}^j]^n$ of the $j$-th Wilson-loop eigenvector. Figure 1c shows the Wannier bands for clockwise configuration with $\Delta d_1 = 0.25$. There are four gapped wannier bands labeled 1 to 4 that come in pairs above and below $v_y = 0$, indicating the vanishing total dipole moment and providing a vital piece to the emergence of the quadrupole topological phase. Noteworthy, the rich Wannier bands here can yield different combinations. The combinations "1 + 3" and "2 + 4" are presented in Fig. 1c, exhibiting the gapped composite Wannier bands. The vanishing total dipole moment can also be captured within the framework of symmetry-indicator invariant[36], i.e., comparing the symmetry representations of the mode profile for occupied bands at high symmetric points. Under the framework of symmetry-indicator invariant with $C_4$ symmetry, bulk dipole moment is expressed as $\mathbf{P}^{(4)} = \frac{1}{2}[X_1^{(2)}](\mathbf{a}_1 + \mathbf{a}_2)$, in which the $C_2$ invariants $[X_1^{(2)}] = \#X_1^{(2)} - \#\Gamma_1^{(2)}$ describes the difference in the number of occupied bands with a given symmetry eigenvalue for X and Γ points[36], and $\mathbf{a}_1$ and $\mathbf{a}_2$ are lattice vectors. The $C_2$ eigenvalues ($\pm 1$) at the high-symmetry points are labeled in Fig. 1b. Since the zero-frequency optical mode is generally even-parity, $\mathbf{P}^{(4)} = (0, 0)$, which is consistent with the results obtained from the Wannier band analysis. Distinguishing the quadrupole topology goes beyond the framework of the symmetry-indicator invariants and instead requires to be diagnosed through the nested Wannier bands. To further describe the topological feature for the quadrupole topology, Fig. 1d shows the $k_y$-resolved polarizations $p_x^{v_y}(k_y)$ of Wannier bands. For Wannier sectors "1 + 3" and "2 + 4" (orange lines in Fig. 1c), they exhibit nontrivial polarization of $\pm 1/2$ due to the glide symmetries constraints on Wannier band polarizations (Supplementary Sec. S1). Moreover, the equivalent polarization in orthogonal direction $p_y^{v_x} = p_x^{v_y}$ can be deduced via $C_4$ symmetry, thus the quantized quadrupole moment is $q_{xy} = 2p_y^{v_x} p_x^{v_y} = 1/2$. Such quadrupole topology revealed by gapped composite Wannier bands is so-called anomalous quadrupole topology to differentiate it from conventional quadrupole topology[30,42,43]. Meanwhile, the nontrivial quadrupole topological phases also occur in counterclockwise configuration, but they are topologically distinguished from clockwise cases by the mutually flipped signs of the Wannier band polarization shown in Fig. 1e and indicate the phase transition at $\Delta d_1 = 0$. These results reveal that the evolution of defects in different directions in our scheme hosts distinct branches of quadrupole topology.

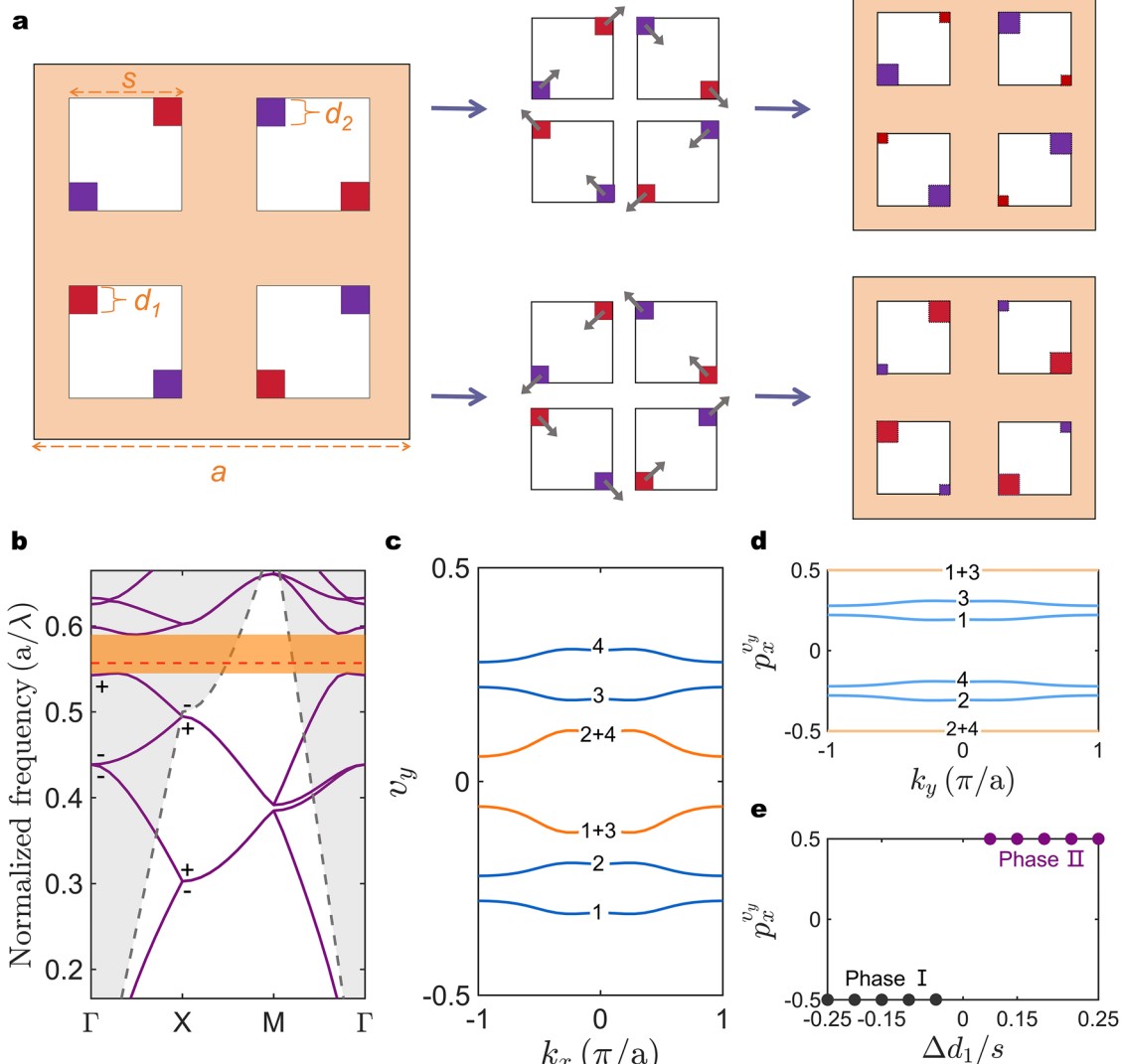

**Fig. 1 | Concept of the photonic quadrupole topological phase under defect evolution. a** Illustration of the defect evolution path in PhC unit cell. **b** Calculated TE-like photonic band structure for $\Delta d_1 = 0.25$, $a = 855$ nm, and $s = 0.3a$. The gray and yellow regions represent the light line and bandgap, respectively. The red dashed line indicates the frequency of corner state. The $C_2$ eigenvalue at $\Gamma$ and $X$ is labeled as "$\pm$". **c** Wannier bands (blue lines) and their combinations (orange lines) for the quadrupole band gap at $\Delta d_1 = 0.25$. **d** The nested Wannier bands for $\Delta d_1 = 0.25$. **e** Evolution of nested Wannier bands under the defect-related parameter $\Delta d_1$, the positive and negative signs of $\Delta d_1$ correspond to clockwise and counter-clockwise directions of evolution, respectively.

## Experimental demonstration of defect-evolved quadrupole topological nanolasers

In photonic HOTIs, a finite-size photonic system constructed by two topologically distinguished PhCs is known to support the extremely localized 0D topological corner state. To investigate the feasibility of the proposed architecture for lasing emission, the defect-evolved clockwise and counterclockwise configuration PhCs are combined to form a quadrupole higher-order topological PhC nanocavity based on previous analyses. Figure 2a shows a schematic of the defect-evolved quadrupole topological nanolaser, in which the different topological regions are identified by green and purple, respectively. The device is fabricated using the 260-nm-thick slab with InGaAsP multiple quantum wells as gain medium[44]. The scanning electron microscope (SEM) image of the fabricated device with $\Delta d_1 = 0.25$ is shown in Fig. 2b, which is composed of two topologically distinguished PhCs, with the left-bottom green region corresponding to the clockwise configuration, while the remaining region is composed of the counterclockwise counterpart that evolves in the opposite direction. Figure 2c further shows a zoomed-in SEM image around the central corner boundary

region. Before numerically studying the corner state, we first verified the edge states at the topologically distinct domains in ribbon-like supercells. Figure 2d shows the projected band structure that reveals the 1D edge states (purple lines) inside the bulk gap. The field distribution for the typical lower frequency edge state is plotted in the top panel of Fig. 2d. The electromagnetic field is strongly localized at the topologically distinct domain. Topological 1D edge state lasing emission from the flat interface can be realized and has been experimentally verified, as shown in Supplementary Sec. S2. To demonstrate the existence of the corner state, Fig. 2e displays the calculated eigen-frequency spectra of the topological nanocavity for a typical magnitude of $\Delta d_1 = 0.25$. A corner state as a single state (orange dot) emerges within the bandgap. The corresponding simulated electric field profile is presented in the inset of Fig. 2e. As expected, the defect-evolved higher-order corner state in the quadrupole topological phase is tightly localized around the corner region assembled by two topologically distinguished slabs. Additionally, this extreme case of defect evolution ($\Delta d_1 = 0.25$) is similar to quadrupole topological structure in acoustic systems with complex arch-shaped geometry

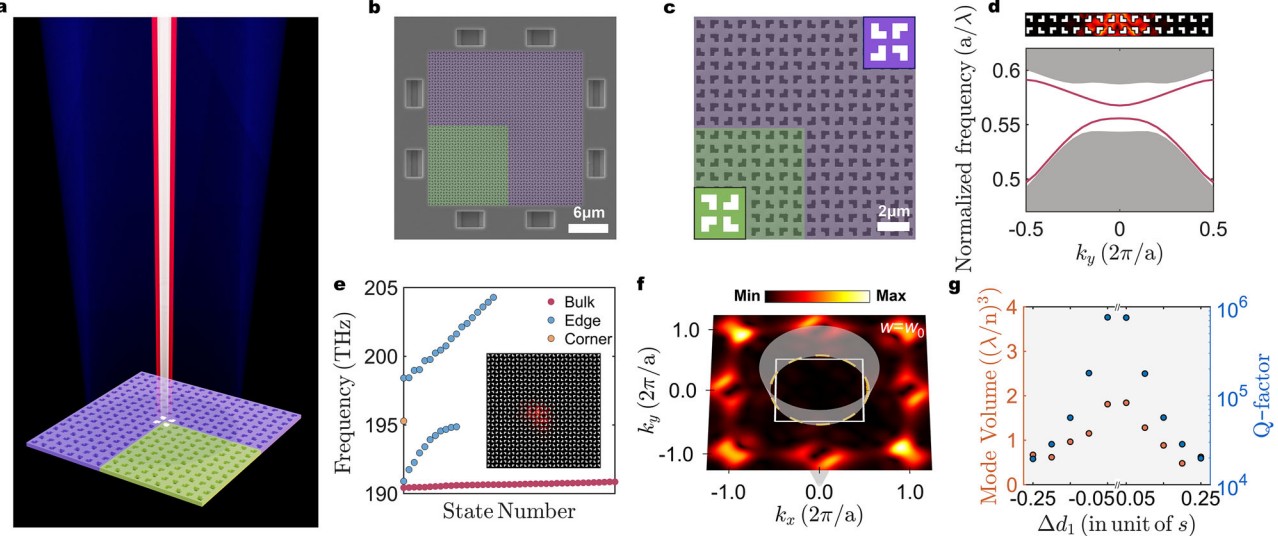

**Fig. 2 | Quadrupole topological nanolaser cavity design. a** Conceptual illustration of proposed defect-evolved quadrupole higher-order topological PhC nanolaser. **b** The top-view SEM image of the fabricated nanolaser with $a = 855$ nm, $s = 0.3a$, and $\Delta d_1 = 0.25$. **c** The zoomed-in SEM image around the central corner region. **d** Projected band structure and the simulated electric field profiles ($|E|^2$) of the edge state. **e** Eigenfrequency spectra of the investigated cavity. The gray regions represent the range of bulk states, and the inset shows the simulated $|E|^2$

profile of the topological corner state with $\Delta d_1 = 0.25$. **f** Spatial Fourier transformation for the corner state. The white square represents the first Brillouin zone. The gray region represents the light-cone boundary, with the dashed yellow circle representing the cross section of the light-cone boundary at the frequency of the corner state. **g** Simulated $V_m$ and $Q$-factor of the topological corner state with different $\Delta d_1$.

transformation[42,43]. Figure 2f shows the spatial Fourier transforms of the $H_z$ field for corner state at the corresponding frequency. The wavevector component inside the light cone boundary (labeled by gray region) is negligible compared to the outside, which is concentrated on the extended Brillouin zone due to being built on the folded band. While the chiral symmetry for most higher-order topological insulators leads to the corner states pinned to zero energy, the chiral symmetry is always broken in photonic crystals, and crystalline symmetries generally do not inherently constrain the energy of corner states. In Supplementary Sec. S3, we show the robustness of the corner state against several defect types and disorders. To further analyze the confined corner state, the parametric dependence of the calculated $V_m$ and quality factors ($Q$-factors) on the defect-related parameter $\Delta d_1$ is shown in Fig. 2g, with $\Delta d_1$ here defined from left-bottom region, while the rest changes in the opposite direction by the same magnitude. The $V_m$ increase as the magnitude of $\Delta d_1$ decreases can be attributed to the gradual narrowing of the bandgap (Supplementary Sec. S4), and gradually shows the trend of the increasing $Q$-factors at the expense of sacrificing the $V_m$. The $Q$-factors are on the order of $10^4$ to near $10^6$, with the out-of-plane loss significantly suppressed, in which the larger $Q$-factors for the case with expanded mode volumes can be attributed to the suppression of out-of-plane loss[45], which is indicated by the smaller wavevector component in the light cone (Supplementary Sec. S5).

The lasing emission of the fabricated devices was characterized by the micro-photoluminescence ($\mu$-PL) system at room temperature. The quadrupole topological nanolasers are optically pumped using a 632 nm pulsed laser with a repetition rate of 200 kHz and a duty cycle of 0.5%. Figure 3a presents the power-dependent spectra of a quadrupole topological PhC nanolaser with structural parameters $a = 855$ nm, $s = 0.3a$, and $\Delta d_1 = 0.25$, measured by locating the optical excitation spot at the corner region (Supplementary Sec. S6). The corner state exhibits suppressed radiation loss, a high $Q$-factor with a small $V_m$, and robustness against defects and disorders, thereby enabling room temperature lasing emission in our scheme and naturally tends to stable single-mode operation. As illustrated in the figure, a sharp lasing peak (~1567 nm) springs from the background of spontaneous emission. The intensity ascends as the input power increases,

exhibiting the stable single-mode operation over a broad range of input powers. And the collected topological corner state nanolaser is linearly polarized, illustrated by the polarization-angle-dependent output intensity (Supplementary Sec. S7). To further demonstrate the signatures of the lasing operation, the corresponding light-in-light-out ($L$-$L$) curve and the linewidth evolution of the device under average input power are illustrated in Fig. 3b, both an obvious kink in the $L$-$L$ curve and the linewidth narrowing effect are observed, presenting clear evidence for the stimulated emission in the fabricated quadrupole topological nanolaser. A relatively low lasing threshold, approximately 0.5 $\mu$W (peak power 100 $\mu$W, power density 3.18 kW/cm$^2$), is estimated from the $L$-$L$ curve, and the linewidth near the threshold of ~0.68 nm corresponds to an experimental $Q$-factor of 2300. In contrast to the simulated $Q$-factor, the lower experimental $Q$-factor may be attributed to the unavoidable fabrication induced imperfections. For the designed quadrupole topological nanolaser, the stimulated emission from the corner state can be further verified by collecting near-field optical profiles using an InGaAs camera. Figure 3c depicts the near-field optical profile measured of a fabricated corner state nanolaser measured above lasing thresholds. The optical mode is spatially localized in the central corner region, and speckle pattern occurs due to the coherent emission above the threshold. Furthermore, the thermal stability for the proposed defect-evolved topological corner state nanolaser is evaluated by placing the devices on a thermoelectric cooler (TEC) stage. The fabricated device presents stable single-mode emission as temperature increases from 30 to 70 °C (Supplementary Fig. S11), suggesting the potential practical applications for the quadrupole topological nanolaser in relatively high temperature scenes. Meanwhile, the quadrupole topological nanolaser is optically pumped at room temperature with various pulse widths ranging from 50 to 150 ns (Supplementary Fig. S12). Higher lasing thresholds are observed at elevated temperatures and with longer pulse widths, mainly due to reduced gain and increased nonradiative recombination rates[46]. To excite the lasing from edge and bulk states, the optical excitation spot is selectively located at the corresponding edge and bulk region, respectively. Figure 3d shows the lasing spectra for different states (the near-field optical profiles and $L$-$L$ curves of edge and bulk states are

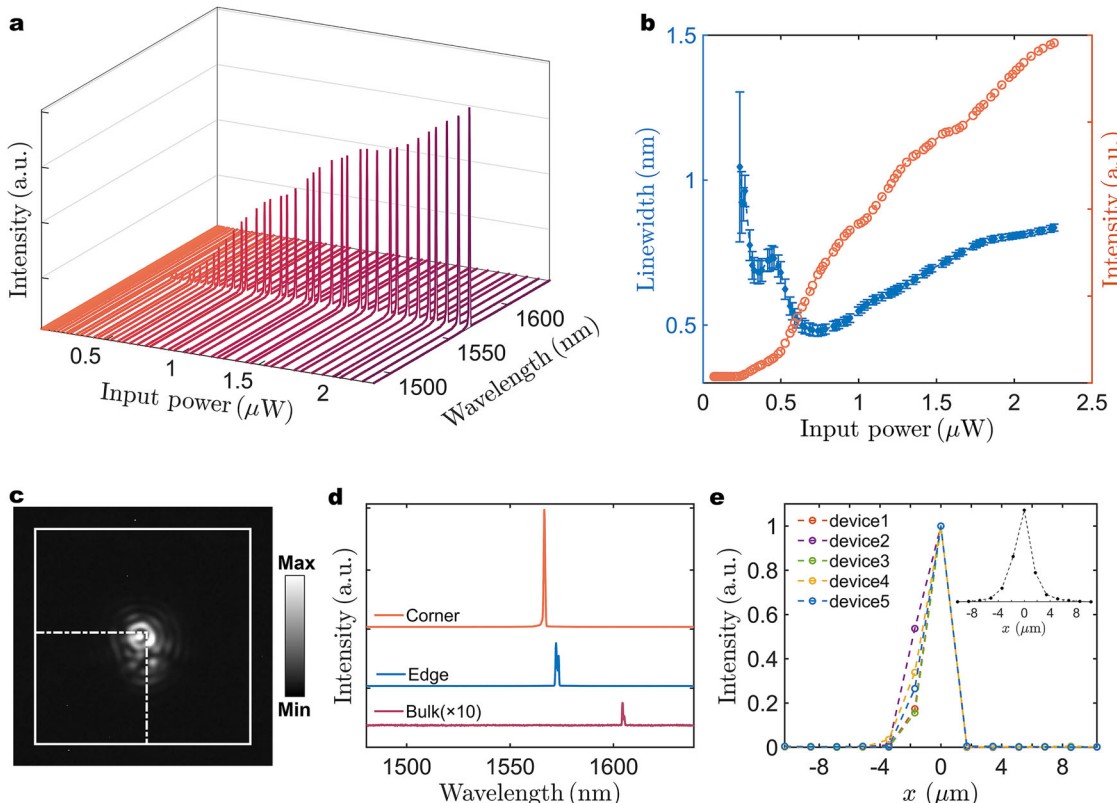

**Fig. 3 | Optical characterizations of the quadrupole higher-order topological nanolasers. a** Measured power-dependent emission spectra of a quadrupole topological nanolaser. **b** *L-L* curve (orange dots) and linewidth (blue dots) under various average input powers. The error bars represent the standard errors deduced by fitting. **c** The near-field optical profiles of corner above the lasing threshold. **d** Measured lasing spectra from the corner, edge, and bulk states, respectively. The intensity from the bulk is magnified 10 times for clarity. **e** Position-dependent normalized PL intensities of five fabricated devices. The negative and positive axis directions correspond to edge region and the topologically distinct region, respectively. The inset shows the simulated electric field distribution above the slab in horizontal direction.

shown in Supplementary Sec. S9). The lasing behavior, including the anticipated topological mode selection and spectral trends, is consistent with the expected characteristics of the corner, edge, and bulk states. To further assess the underlying field profiles of the corner state, we examine the position-dependent PL intensities shown in Fig. 3e. The observed lasing mode is tightly localized in the corner region and exhibits a slower decay along the edge region direction compared to the opposite direction (the topologically distinct region), which is consistent with the simulated results. Moreover, the far-field pattern of corner state lasing emission exhibits a unique distribution, distinct from that of edge state (Supplementary Fig. S14). These results, together with the following tunable wavelength emission, unambiguously confirm the single-mode lasing operation origin from the designed topological corner state.

In addition to the realization of photonics quadrupole phase transition, the defect evolution also provides an opportunity for the topological nanolasers with spectrally tunable emission. Distinct from the previous prevalent wavelength manipulation strategy[34,47], the tunable wavelength emission of quadrupole topological nanolasers here is realized only through manipulating the defect-related parameter $\Delta d_1$ (Fig. 4a). To obtain a quantitative insight into the modulation of the output lasing wavelength upon the defect evolution, the variation of simulated wavelength (blue dots) of the corner state with different $\Delta d_1$ is shown in Fig. 4b (geometric parameter $s = 0.287a$ here to better match the experimental lasing wavelengths). It can be seen that the wavelength of the corner state increases with the magnitude of $\Delta d_1$. For the fabricated nanolasers, the lasing wavelengths with the defect-related parameter $\Delta d_1$ are also presented in Fig. 4b (orange dots). Despite irregularities and roughness of the devices induced by

fabrication fluctuations, the lasing wavelengths exhibit a systematic redshift that strongly correlates with increasing $\Delta d_1$, closely aligning with the simulation trend. Figure 4c further shows the corresponding normalized lasing spectra, exhibiting that the lasing wavelengths are approximately tuned from 1519 to 1543 nm, which offers a tunable wavelength range of 24 nm (The individual lasing spectra, *L-L* curve, and linewidth for the nanolasers with representative defect parameter $\Delta d_1 = 0.18$ to 0.15 are shown in Supplementary Fig. S16). These results reveal that the geometrical defect evolution can be employed as an efficient method to achieve both precise lasing wavelength tuning of corner state and quadrupole topological phase. This approach establishes a versatile design strategy for wavelength-tunable topological nanolaser, particularly advantageous for high-throughput densely integrated photonic systems.

## Discussion
Our study demonstrates a scheme for realizing the photonic higher-order corner state in quadrupole topological phase with introduced defect evolution as a degree of freedom and validates them in semiconductor nanolaser. The proposed quadrupole topological nanolaser demonstrates stable single-mode operation in the telecommunication C-band at room temperature, achieving an ultralow threshold of 0.5 μW and an experimental *Q*-factor of 2300, while maintaining robust performance up to 70 °C. The lasing behavior remains sustained even under introduced structural perturbations. The corresponding tunable lasing emission is further verified by manipulating the evolution of geometrical defects. These quadrupole topological nanolasers presented here open prospects for the topological on-chip nanoscale coherent light sources. Meanwhile, the corner state in the quadrupole

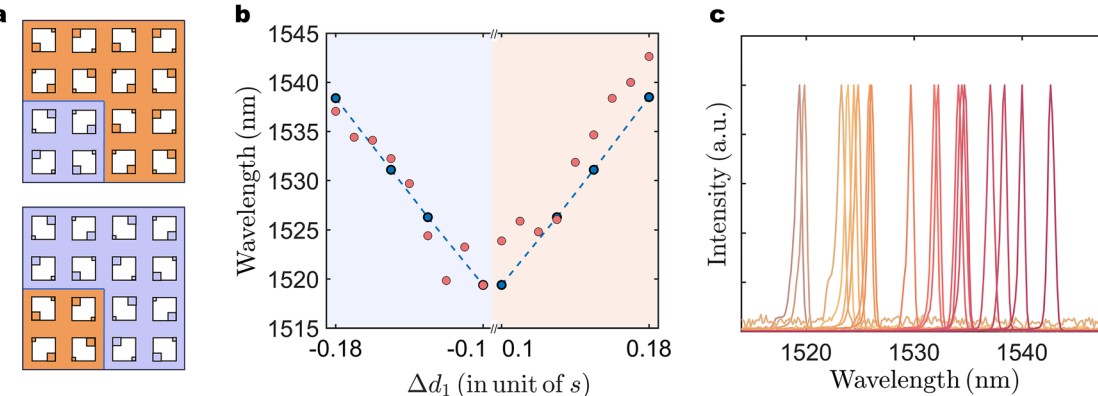

**Fig. 4 | Tunable wavelength manipulation under defect evolution. a** Schematic representation of defect evolution. **b** Simulated resonant wavelength and experimental lasing wavelength of the fabricated quadrupole topological nanolasers by varying $\Delta d_1$. The configuration of the minus/plus sign of $\Delta d_1$ corresponds to the top/bottom panel in Fig. 4a. **c** Normalized measured lasing spectra from nanolasers under defect evolution.

topological phase by strategically modulating the defect evolution also provides inspiration for exploring the fundamental physics and practical applications with intriguing topological properties. Many possible extensions could be envisioned. For example, such topological corner state with defect evolution may find applications in wavelength-division multiplexing[48] and reconfigurable imaging[49,50], and it may also be able to combine with the quantum emitter for the purpose of single photon generation in topology-driven nanophotonic devices[51]. Moreover, our results might bring about new opportunities to investigate topological rainbow trapping[44,52,53], large-area topological corner states[54], and exploring the influence of defect evolution under different topological phases may offer a possible route toward rich topological states. While the photonics quadrupole topological phase demonstrated here is based on a specific photonic system, we envisage that the proposed scheme could be generalized to other platforms, such as microwaves[55,56] and thermal systems[57], to further explore fascinating topological phenomena.

## Methods
### Device fabrication
A 260-nm-thick slab with six strained InGaAsP multi-quantum wells as the gain medium was used to fabricate the defect-evolved quadrupole topological nanolasers. First, plasma-enhanced chemical vapor deposition was employed to deposit 90-nm-thick SiO$_2$ as a hard mask. Then, the designed topological PhC structure was defined using electron beam lithography in photoresist of PMMA950 A4 and transferred to the SiO$_2$ hard mask and further into the active layer by plasma dry etching. Afterwards, the residual resist was removed using O$_2$ plasma, and the buffered oxide etching solution was used to remove the remaining SiO$_2$ hard mask. Finally, the InP sacrificial layer is selectively removed by diluted hydrochloric acid solution, resulting in the suspended defect-evolved quadrupole topological nanolasers.

### Optical measurement
The defect-evolved quadrupole topological nanolasers were optically pumped using a 632 nm nanosecond pulsed laser with a repetition rate of 200 kHz in the $\mu$-PL system. The pumping beam, with a spot diameter of approximately 2 $\mu$m, was focused onto the sample using a 100× objective lens and spatially controlled by piezoelectric nanopositioners. The emission from the fabricated nanolaser was collected by the same objective lens and characterized by the spectrometer with an infrared InGaAs detector cooled by liquid nitrogen. The near-field optical images were captured by using an InGaAs camera, and the far-field pattern was obtained through a 4 f optical system. The time-resolved photoluminescence measurements were performed using a time-correlated single-photon counting system.

### Numerical simulations
In numerical studies, we conduct 3D simulations to obtain the photonic bulk and projected band structure, eigenfrequency spectra, $Q$-factor, $V_m$, and field profile by the finite element method. In the 3D simulations, the refractive index of the gain slab is set to $n = 3.25$. To compute the Wannier band and nested Wannier bands, 2D simulations are performed with an effective refractive index of $n = 2.6$. The mode volume $V_m$ of the topological nanocavity is calculated by $V_m = \frac{\int \varepsilon(\mathbf{r})|E(\mathbf{r})|^2 dV}{max(\varepsilon(\mathbf{r})|E(\mathbf{r})|^2)}$, where $\varepsilon(\mathbf{r})$ and $E(\mathbf{r})$ denote the spatially dependent dielectric constant and electric field intensity, respectively.

### Reporting summary
Further information on research design is available in the Nature Portfolio Reporting Summary linked to this article.

## Data availability
The data supporting the findings of this study are available within the article and its supplementary file. Source data are provided with this paper.

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

## Acknowledgements

This work is supported by the National Natural Science Foundation of China (Grant No. 62304080 (T.Z.)), the Guangdong Basic and Applied Basic Research Foundation (Grant No. 2024A1515010802 (T.Z.)), the Science and Technology Projects in Guangzhou (Grant No. 2024A04J3683 (T.Z.)), the Fundamental Research Funds for the Central Universities (Grant No. 2023ZYGXZR068 (T.Z.)). T.Z. acknowledges the startup funds from South China University of Technology. The authors acknowledge the support from the Micro & Nano Electronics Platform (MNEP) of SCUT for device fabrication and characterization, and the Micro/Nano Fabrication Platform of the Institute of Semiconductors, Guangdong Academy of Sciences, for support in device fabrication.

## Author contributions

The concepts were developed by S.G. and T.Z. S.G. performed the numerical calculations and device simulations. W.H. and F.T. fabricated the devices. S.G. and Y.Z. carried out the optical measurements with assistant from Y.W. S.G. wrote the paper with input from all coauthors. T.Z. supervised the project.

## Competing interests

The authors declare no competing interests.
