## [Transparent Peer Review file · Nature Communications]

Defect-evolved quadrupole higher-order topological nanolasers

Corresponding Author: Professor Taojie Zhou

Version 0:

Reviewer comments:

Reviewer #1

(Remarks to the Author)

The manuscript presents a method to generate quadrupole corner states via "defect evolution" and demonstrates a topological nanolaser based on these states. This manuscript is well-written and organized. While the concept of defect evolution for topological phase engineering is novel and theoretically sound, the experimental demonstration of the topological laser lacks sufficient novelty compared to prior experimental works (e.g., the authors' own study in ACS Photonics 9, 3824 (2022)). My main concern is the absence of a clear comparison between the defect-evolved quadrupole corner state and conventional higher-order topological corner states, which undermines the significance of the approach. Why is this method necessary and better than other higher-order topological lasers? The authors should clarify this concern first.

Below are some other minor comments:

1. The authors define their method as "defect evolution," but the final photonic crystal structure is periodic without obvious defects. It is unclear how the term "defect" is justified here, as the design involves systematic geometric adjustments rather than introducing disorder or missing elements. Please clarify the definition of "defect" in this context.
2. The caption states, "The gray line and yellow region represent the light line and bandgap, respectively," but the visual elements in Fig. 1a do not clearly align with this description. Please check.
3. Quadrupole topological phases in magnetic photonic crystals (e.g., Nat. Commun. 11, 3119 (2020); Natl. Sci. Rev. 11, nwa121 (2024)) rely on magneto-optical effects, whereas this work achieves quadrupole topology without magnetism. What is the fundamental difference between those two methods? Please explicitly compare them.
4. The manuscript states that Q-factors increase as the mode volume (V_m) expands, which contradicts the conventional understanding that tighter light confinement (smaller V_m) enhances Q-factors. A more detailed explanation is needed.
5. The optical pumping setup lacks clarity such as what is the pump spot size and its spatial overlap with the corner state.
6. In Fig. 3b, the light-in-light-out (L-L) curve exhibits an indistinct threshold kink when pumping power increases. Can the authors explain why?
7. The temperature-dependent lasing spectra (Fig. 3e) shows no evidence of bulk modes at all. Can the authors explain why?

Reviewer #2

(Remarks to the Author)

Reviewer #3

(Remarks to the Author)

The development of quadrupole topological lasers is a timely and important research topic. The idea of defect evolution to realize quadrupole phases could potentially enrich the field of topology-based nanophotonics. However, while the lasing

operation in experiment itself appears convincing, I find that the topological origin of the observed lasing remains insufficiently demonstrated. At present, the results could be interpreted as stemming from other conventional localized modes rather than quadrupole topological states.

The suggested structure in the manuscript still raises a specific concerns, such as chiral symmetry, robustness, and quadrupole-based lasing. Related to the comment above, I have some comments for clarification.

Comment 1.

The manuscript lacks a clear explanation of where the quadrupole topological mode emerges within the photonic band structure. Specifically, it is essential to identify whether the edge and corner modes lie inside or outside the light cone. Given that modes inside the light cone are susceptible to significant radiation losses, understanding their position is crucial, especially in the context of laser performance. The authors are encouraged to present the band structure overlaid with the light cone and to indicate the frequency and momentum location of the corner modes, preferably in relation to the gap illustrated in Fig. 1d.

Comment 2.

To verify a topological transition driven by the defect parameter Δd , it would be helpful to track the evolution of the photonic band structure as a function of Δd . The authors need to perform such a scan and to visualize the modal evolution, particularly at high-symmetry points such as the M point. Including mode profiles and phase distributions (e.g., $\arg(E_{\{z\}})$) for both trivial (supercell, $\Delta d=0$) and topological phases would help clarify whether band inversion or gap opening occurs. At present, the distinction between trivial and topological phases is not clearly visible in the data.

Comment 3.

It is not clear whether the simulations shown in Fig. 1b, Fig. 2d, and Fig. 2e were performed using a 2D approximation or full 3D finite-element calculations. Given the slab nature of the experimental system, a thickness is critical for accurate modeling of mode confinement, radiation losses, and Q factor.

Comment 4.

While the authors mention the robustness of the corner state against disorder (see Supplementary Information), it remains unclear whether the structure indeed exhibits genuine topological protection. In many topological systems, such protection arises from underlying chiral or crystalline symmetries. The manuscript should address whether the corner states are pinned to midgap via symmetry constraints or are simply localized modes that are weakly perturbed by disorder. If topological protection is claimed, a clearer explanation of the protecting symmetry and the nature of the topological invariant is needed.

Comment 5.

The Q-factor plotted in Fig. 2e shows an inverse relationship with Δd_1 , with the highest Q obtained as Δd approaches zero (=the trivial lattice limit). This observation raises the question whether the trivial design may, in fact, be more favorable for laser operation. If no protection mechanisms exist to ensure robustness or midgap pinning, and if the topological design results in lower Q, the motivation for using a quadrupole lattice in a laser becomes unclear. The authors should comment on this trade-off and explain whether the quadrupole configuration offers any definitive advantage for conceptual novelty.

Comment 6.

The manuscript lacks simulations or experimental discussion of edge states. In higher-order topological insulators, the edge states is often essential in confirming the hierarchy of bulk-edge-corner correspondence. The authors are encouraged to simulate or measure the edge configurations between topologically distinct domains. Simulations at 1D interfaces (edges) would help resolve this issue.

Comment 7.

To better understand the modal characteristics, I suggest the authors simulate the far-field patterns and analysis their polarization dependence for the simulated lasing mode. Quadrupole states might be expected to exhibit unique field symmetries and polarization features. These could help distinguish them from other possible defect-localized modes. Providing polarization-resolved near-field or far-field measurements would also be valuable.

Comment 8.

Related to the previous point, it would be helpful to show polarization-resolved near-field images of the lasing emission and to compare these with the simulated quadrupole mode profiles. I hope that such measurements would offer further insight into whether the lasing mode indeed arises from the expected quadrupole corner state. Additionally, further verification is needed to ensure that the observed lasing mode is not a fundamental mode (1st band) arising from non-topological band folding or structural asymmetry.

Comment 9.

It is unclear how temperature-dependent measurements (Fig. 3f, 3g) relate to the topological aspects of the design. Is there any reason to expect a topological invariant or corner-state localization to degrade (or persist) with temperature? To my knowledge, it is confusing why the authors showed the thermal dependence for quadrupole lasing operations.

Comment 10.

For further insight into the lasing behavior, the authors might consider varying the duty cycle of the pump pulses by increasing the pulse width at fixed repetition rate. This would help assess thermal effects, and potentially the CW response of the lasing mode. While not essential, such measurements could enhance the depth of characterization.

While the reported results are promising in terms of device functionality and experimental feasibility for lasing, the connection between lasing and quadrupole topology is not sufficiently substantiated at this stage. I encourage the authors to address the above points (notably, concerning the topological origin of the lasing mode). Solving those clarifications, I will look forward a definitive demonstration for a next round of revision.

Version 1:

Reviewer comments:

Reviewer #1

(Remarks to the Author)

I appreciate the authors' efforts during the revision. The previous technical concerns have been resolved. The authors have introduced a novel approach for constructing a photonic quadrupole insulator. This method does not rely on tight-binding models such as the 2D SSH model, nor does it require magnetic materials, making it highly promising for photonic applications such as topological lasers as demonstrated in this work. I recommend the manuscript for publication.

Reviewer #3

(Remarks to the Author)

This manuscript introduces an topologically quadrupolar design for topological nanolasers. However, I still have several key concerns:

- (1) The position of the mode relative to the light cone is not clearly resolved.
- (2) The connection between the simulated Wannier-like topology and the lasing mode remains indirect.
- (3) I am not sure that the experimental evidence does not yet clearly support the interpretation of a quadrupole HOTI mode.

I appreciate the authors' efforts for high quality experimental results. However, the origin of quadrupolar topology and their experimental lasing system remains insufficiently demonstrated. Below, I provide detailed comments for addressing my concerns.

Comment 1.

Most central and critical issue is that the claimed quadrupole HOTI mode seems to form a frequencies in above-light cone regime. While Wannier-like photonic system may be allowed in theory or idealized simulations, but in practice, light cone and its position in photonic bands leads to substantial radiation loss, and limit the Q factor and making laser operation extremely challenging.

Most successful demonstrations of photonic topological lasers operate near the fundamental band and their topology because it is clearly positioned below to the light cone. In contrast, the current design places the quadrupole mode well above the light cone at Gamma and X. This reason makes it difficult whether the observed lasing is truly from a topological mode, or a conventional cavity mode. The authors should clearly identify the wavevector and frequency of the quadrupole mode relative to the light cone in the band diagram

Comment 2.

If the mode lies inside the light cone, it becomes important to clarify which wavevector the red dashed line (corner state) in Fig. 1b corresponds to. Specifically, at which momentum point is the band topology formed?

Comment 3.

Although robustness tests are presented (Fig. S3), the perturbations are too far from the corner. This does not adequately test whether the corner mode is topologically protected. Generally, it would be meaningful to introduce bulk disorder including or near the corner region.

Comment 4.

As the authors already addressed in their response, the existence of chiral symmetry in current design should be mentioned in the main text. While there are topological studies on systems with chiral symmetry, such investigations remain relevant even when the symmetry is not strictly preserved in photonic system. It is important to reflect this point in the manuscript as well. In my view, even a single sentence clarifying this in the main text would be sufficient for readers.

Comment 5.

The authors explain on Wannier band polarizations and nested Wilson loops to argue for quadrupole topology. I agree to this theoretical approaches to design a quadrupole in photonics and it already introduced in several times in other reports. However, in current version there still lack clear real-space verification that the experimental lasing mode corresponds to the predicted quadrupole corner state.

Comment 6.

The authors are encouraged to provide unsaturated real-space images of the lasing mode above threshold. To confirm that the observed lasing mode indeed corresponds to a Wannier-based quadrupole topological state, it is essential to analyze the spatial symmetry and vector-field distribution of the mode.

Currently, the mode profile images (Fig. 3c–e) do not provide sufficient information to assess the underlying field profiles, due to the saturation. It is not possible to analyze the in-plane E-field components, which may introduce evidence for identifying the mode's symmetry and matching to theoretical quadrupole corner states model.

Comment 7.

The reported threshold ($\sim 0.5 \mu\text{W}$) appears extremely low and raises concerns about how it was measured. Is the value based on average or peak pump power? This must be clarified in both the text and figure captions. If using average power, convert to peak power for comparison with other nanolaser works. On general understanding, a peak pump power of $0.5 \mu\text{W}$ is extremely low, and it is unlikely that suggested III–V semiconductors would respond at such levels.

Comment 8.

I did not specifically intend to highlight the M point alone, but rather to suggest that the authors identify the conditions under which topologically quadrupole mode inversion occurs, and the corresponding wavevector where it takes place. I mentioned the M point simply as an example, because a mode inversion occurring within the light cone is more likely to result in a physically realizable lasing system.

The current analysis still lacks sufficient mode profile information of the unit cell as a function of wavevector. This makes it difficult to determine under what conditions the E-field distribution can be considered as “Wannier-like”.

Upon reviewing the current data, however, the closest relevant bands seem to lie along the Γ –X direction. I still find it unclear whether the observed lasing mode genuinely originates from a topological Wannier-like system, and whether this system can be convincingly interpreted as a laser based on a quadrupole topological phase.

Version 2:

Reviewer comments:

Reviewer #3

(Remarks to the Author)

The authors have carefully and substantially addressed my previous concerns and significantly improved the revised manuscript. In particular, the revision clarifies

- (1) the position of the lasing mode relative to the light cone,
- (2) the relationship between the Wannier-based description and the quadrupole topological phase,
- (3) the experimental evidence supporting the lasing interpretation, and
- (4) the symmetry considerations, including the role and limitations of chiral symmetry in photonic systems.

Overall, I am satisfied that the central claims of the manuscript are now adequately supported, and I believe that the work

has reached the level appropriate for publication in Nature Communications.

REVIEWER COMMENTS

Reviewer #1 (Remarks to the Author):

The manuscript presents a method to generate quadrupole corner states via "defect evolution" and demonstrates a topological nanolaser based on these states. This manuscript is well-written and organized. While the concept of defect evolution for topological phase engineering is novel and theoretically sound, the experimental demonstration of the topological laser lacks sufficient novelty compared to prior experimental works (e.g., the authors' own study in ACS Photonics 9, 3824 (2022)). My main concern is the absence of a clear comparison between the defect-evolved quadrupole corner state and conventional higher-order topological corner states, which undermines the significance of the approach. Why is this method necessary and better than other higher-order topological lasers? The authors should clarify this concern first.

Below are some other minor comments:

1. The authors define their method as "defect evolution," but the final photonic crystal structure is periodic without obvious defects. It is unclear how the term "defect" is justified here, as the design involves systematic geometric adjustments rather than introducing disorder or missing elements. Please clarify the definition of "defect" in this context.
2. The caption states, "The gray line and yellow region represent the light line and bandgap, respectively," but the visual elements in Fig. 1a do not clearly align with this description. Please check.
3. Quadrupole topological phases in magnetic photonic crystals (e.g., Nat. Commun. 11, 3119 (2020); Natl. Sci. Rev. 11, nwae121 (2024)) rely on magneto-optical effects, whereas this work achieves quadrupole topology without magnetism. What is the fundamental difference between those two methods? Please explicitly compare them.
4. The manuscript states that Q-factors increase as the mode volume (V_m) expands, which contradicts the conventional understanding that tighter light confinement (smaller V_m) enhances Q-factors. A more detailed explanation is needed.
5. The optical pumping setup lacks clarity such as what is the pump spot size and its spatial overlap with the corner state.
6. In Fig. 3b, the light-in-light-out (L-L) curve exhibits an indistinct threshold kink when pumping power increases. Can the authors explain why?
7. The temperature-dependent lasing spectra (Fig. 3e) shows no evidence of bulk modes at all. Can the authors explain why?

Reviewer #2 (Remarks to the Author):

Reviewer #3 (Remarks to the Author):

The development of quadrupole topological lasers is a timely and important research topic. The idea of defect evolution to realize quadrupole phases could potentially enrich the field of topology-based nanophotonics. However, while the lasing operation in experiment itself appears convincing, I find that the topological origin of the observed lasing remains insufficiently demonstrated. At present, the results could be interpreted as stemming from other conventional localized modes rather than quadrupole topological states.

The suggested structure in the manuscript still raises a specific concerns, such as chiral symmetry, robustness, and quadrupole-based lasing. Related to the comment above, I have some comments for clarification.

Comment 1.

The manuscript lacks a clear explanation of where the quadrupole topological mode emerges within the photonic band structure. Specifically, it is essential to identify whether the edge and corner modes lie inside or outside the light cone. Given that modes inside the light cone are susceptible to significant radiation losses, understanding their position is crucial, especially in the context of laser performance. The authors are encouraged to present the band structure overlaid with the light cone and to indicate the frequency and momentum location of the corner modes, preferably in relation to the gap illustrated in Fig. 1d.

Comment 2.

To verify a topological transition driven by the defect parameter Δ_d , it would be helpful to track the evolution of the photonic band structure as a function of Δ_d . The authors need to perform such a scan and to visualize the modal evolution, particularly at high-symmetry points such as the M point. Including mode profiles and phase distributions (e.g., $\arg(E_{\{z\}})$) for both trivial (supercell, $\Delta_d=0$) and topological phases would help clarify whether band inversion or gap opening occurs. At present, the distinction between trivial and topological phases is not clearly visible in the data.

Comment 3.

It is not clear whether the simulations shown in Fig. 1b, Fig. 2d, and Fig. 2e were performed

using a 2D approximation or full 3D finite-element calculations. Given the slab nature of the experimental system, a thickness is critical for accurate modeling of mode confinement, radiation losses, and Q factor.

Comment 4.

While the authors mention the robustness of the corner state against disorder (see Supplementary Information), it remains unclear whether the structure indeed exhibits genuine topological protection. In many topological systems, such protection arises from underlying chiral or crystalline symmetries. The manuscript should address whether the corner states are pinned to midgap via symmetry constraints or are simply localized modes that are weakly perturbed by disorder. If topological protection is claimed, a clearer explanation of the protecting symmetry and the nature of the topological invariant is needed.

Comment 5.

The Q-factor plotted in Fig. 2e shows an inverse relationship with Δd_1 , with the highest Q obtained as Δd approaches zero (=the trivial lattice limit). This observation raises the question whether the trivial design may, in fact, be more favorable for laser operation. If no protection mechanisms exist to ensure robustness or midgap pinning, and if the topological design results in lower Q, the motivation for using a quadrupole lattice in a laser becomes unclear. The authors should comment on this trade-off and explain whether the quadrupole configuration offers any definitive advantage for conceptual novelty.

Comment 6.

The manuscript lacks simulations or experimental discussion of edge states. In higher-order topological insulators, the edge states is often essential in confirming the hierarchy of bulk-edge-corner correspondence. The authors are encouraged to simulate or measure the edge configurations between topologically distinct domains. Simulations at 1D interfaces (edges) would help resolve this issue.

Comment 7.

To better understand the modal characteristics, I suggest the authors simulate the far-field patterns and analysis their polarization dependence for the simulated lasing mode. Quadrupole states might be expected to exhibit unique field symmetries and polarization features. These could help distinguish them from other possible defect-localized modes. Providing polarization-resolved near-field or far-field measurements would also be valuable.

Comment 8.

Related to the previous point, it would be helpful to show polarization-resolved near-field

images of the lasing emission and to compare these with the simulated quadrupole mode profiles. I hope that such measurements would offer further insight into whether the lasing mode indeed arises from the expected quadrupole corner state. Additionally, further verification is needed to ensure that the observed lasing mode is not a fundamental mode(1st band) arising from non-topological band folding or structural asymmetry.

Comment 9.

It is unclear how temperature-dependent measurements (Fig. 3f, 3g) relate to the topological aspects of the design. Is there any reason to expect a topological invariant or corner-state localization to degrade (or persist) with temperature? To my knowledge, it is confusing why the authors showed the thermal dependence for quadrupole lasing operations.

Comment 10.

For further insight into the lasing behavior, the authors might consider varying the duty cycle of the pump pulses by increasing the pulse width at fixed repetition rate. This would help assess thermal effects, and potentially the CW response of the lasing mode. While not essential, such measurements could enhance the depth of characterization.

While the reported results are promising in terms of device functionality and experimental feasibility for lasing, the connection between lasing and quadrupole topology is not sufficiently substantiated at this stage. I encourage the authors to address the above points(notably, concerning the topological origin of the lasing mode). Solving those clarifications, I will look forward a definitive demonstration for a next round of revision.

Reviewer #1

The manuscript presents a method to generate quadrupole corner states via "defect evolution" and demonstrates a topological nanolaser based on these states. This manuscript is well-written and organized. While the concept of defect evolution for topological phase engineering is novel and theoretically sound, the experimental demonstration of the topological laser lacks sufficient novelty compared to prior experimental works (e.g., the authors' own study in ACS Photonics 9, 3824 (2022)). My main concern is the absence of a clear comparison between the defect-evolved quadrupole corner state and conventional higher-order topological corner states, which undermines the significance of the approach. Why is this method necessary and better than other higher-order topological lasers? The authors should clarify this concern first.

Response: We thank the reviewer for the positive recognition that our work is "well-written and organized" and that "the concept of defect evolution for topological phase engineering is novel and theoretically sound". And we are appreciative of the informative comments and suggestions that help us to enhance the quality of this work. Regarding the concerns about the novelty of the experimental demonstration of the topological laser. We would like to clarify these from the following :

In our previous study in ACS Photonics 9, 3824 (2022), we demonstrate corner-state nanolasers grown on a CMOS-compatible Si platform. Such corner states are the signature characteristic of the Wannier-type HOTIs in real space, in which the topological property is generally characterized by 2D Zak phase (bulk polarization) and mainly relies on the difference between inter- and intracell couplings. Instead, the introduction of tailored defect in our work naturally produces a quadrupole band gap. For quadrupole TIs, they host vanishing bulk polarization but a quantized quadrupole moment acting as the topological invariant. Compared with the quadrupole topological phase in our work, the Wannier-type HOTIs belong to a different class of symmetry-protected topological phases, which can be seen as the 2D generalization of the SSH model with 1D Zak phase. The realization of Wannier-type HOTIs in photonic systems can be traced back to early research in microwave platforms [Phys. Rev. B 98, 205147 (2018), Phys. Rev. Lett. 122, 233903 (2019), Phys. Rev. Lett. 122, 233902 (2019)] and subsequently such a concept and manipulation strategy is used to construct topological nanocavities [Optica 6, 786-789 (2019)], thus motivating the applications in topological nanolasers based on Wannier-type HOTIs, e.g., low-threshold quantum dot corner-state nanolasers [Light Sci. Appl. 9, 109 (2020)], the lasing of hierarchical eigenstates [ACS Photonics 7, 2027 (2020)], the coupling between the corner states and their lasing action [Nat Commun 11, 5758 (2020)], and the corner state lasing based on coupled perovskite micropillars [Sci. Adv. 9, eadg4322 (2023)]. Compared to these studies, the key contribution of our work is to propose a defect evolution method for realizing

photonic quadrupole topological phase transition, and the room temperature quadrupole higher-order topological nanolaser is realized for the first time under this method. We noticed that the previous corner state lasers with Q -factor limited to 10^3 to 10^4 , the simulated Q -factor of the proposed nanocavity is as high as 10^4 to near 10^6 without further optimized the structural parameters of the corner, and the lasing threshold is much smaller than the reported quantum-well-based corner state lasers. In the revised version, we emphasize that this is based on different topological phases, and a table is shown below to compare with prior corner state lasers to more clearly highlight the distinctive contributions of our work.

Type	Gain material	Wavelength	Threshold	Experimental Q -factor	Operation conditions	Ref.
2D SSH model	InGaAs quantum dot	~1182nm	~1 μ W	~1700	CW (4.2K)	31
2D SSH model	InGaAsP quantum wells	~1550nm	~400 μ W	/	Pulsed (RT)	32
2D SSH model	InGaAsP quantum wells	~1485nm	~7 kW/cm ²	/	Pulsed (RT)	33
2D SSH model	InAs/GaAs quantum dot	1341nm	~3 μ W	~1960	CW (RT)	34
2D SSH model	CsPbBr ₃	~539nm	10 μ J/cm ²	~900	Pulsed (RT)	35
Twisted PhC	InGaAsP quantum wells	~1497nm	~23 μ W	~714	Pulsed (4K)	36
Defected PhC	InGaAsP quantum wells	~1567nm	~0.5 μ W	~2300	Pulsed (RT)	Our work

Table. S1| Comparison of our defect-evolved quadrupole higher-order topological nanolasers and other corner state lasers.

Meanwhile, the proposed defect evolution method inspired the innovation strategy of the PhC laser-simulated novelty topological state and provides avenues for the further research of topological photonics and photonic devices. In addition to topological lasers, the proposed defect evolution method for topological quadrupole phase may find applications in nonlinear photonics and topological photonic crystal fibers [Light Sci. Appl. 9, 202 (2020)]. And one may use the similar method of defect evolution to explore a plethora of intriguing ideas for further research, which is promising in terms of device functionality and experimental feasibility for lasing, as commented by Reviewer #3.

On the other side, embracing distinct topological physics or the manipulation strategy of light confinement is non-trivial for nanophotonic devices. Fruitful examples include the various topological edge-state lasers based on different types of band topology and topological defect lasers with real-space topology [e.g., *Science* 358, 636 (2017), *Science* 359, eaar4005 (2018), *Phys. Rev. Lett.* 125, 013903 (2020), *Nat. Commun.* 12, 3434 (2021), *Nature* 578, 246 (2020), *Light Sci. Appl.* 9, 127 (2020), *Nat. Photonics* 11, 651 (2017), *Phys. Rev. Lett.* 120, 113901 (2018), *Light Sci. Appl.* 8, 40 (2019), *Nat. Photonics* 16, 279 (2022), *Light Sci. Appl.* 12, 255 (2023), *Nat. Commun.* 14, 707 (2023), *Nat. Commun.* 15, 4431 (2024)], ranging from cryogenic to room temperature, optical to electrical pumping, micro-rings/pillars to nanocavities, and have led to outstanding discoveries such as topological bulk lasers in the quantum spin Hall phase [*Nat. Nanotechnol.* 15, 67 (2020), *Light Sci. Appl.* 12, 145 (2023)] and topological valley polaritons [*Nat. Nanotechnol.* 19, 1283 (2024), *Nat. Commun.* 15, 10563 (2024)]. Other than that, the introduction of the magic angle as a new strategy to manipulate light confinement in laser cavities [*Nat. Nanotechnol.* 16, 1099 (2021)] has also promoted the research of reconfigurable nanolaser arrays [*Nature* 624, 282 (2023)], atomic-scale field localization [*Nature* 632, 287 (2024)], cavity quantum electrodynamics [*Nat. Commun.* 16, 4634 (2025)], and exciton polariton condensation [*Sci. Adv.* 11, eadx2361 (2025)]. The above-mentioned works indicate the importance of different topological physics and manipulation strategies for nanocavity design.

Consequently, we believe that our work provides valuable insights for the fields of topological photonics, nanophotonics, and nanolasers, as well as a possible reference approach for the community to develop further research. We hope the above explanations emphasize the novelty and impact of our work.

1. The authors define their method as "defect evolution," but the final photonic crystal structure is periodic without obvious defects. It is unclear how the term "defect" is justified here, as the design involves systematic geometric adjustments rather than introducing disorder or missing elements. Please clarify the definition of "defect" in this context.

Response: We thank the reviewer for the suggestion. We agree with the reviewer that the final photonic crystal structure is periodic, without disorder or missing elements in real space. The defects in this context refer to the square notch into the air hole, which is typically seen in metasurface design [e.g. *Phys. Rev. Lett.* 123, 253901 (2019)]. To prevent potential confusion or misunderstandings, we have added the following sentences to page 3 in the revised manuscript: "the notch structural defects at the diagonal position for each original air hole are introduced, and the defect evolution in this context refers to the systematic geometric adjustments of the geometrical defects for air holes".

2. The caption states, "The gray line and yellow region represent the light line and bandgap,

respectively," but the visual elements in Fig. 1a do not clearly align with this description. Please check.

Response: We thank the reviewer for pointing out our omission. The error has been corrected in the revised manuscript.

3. Quadrupole topological phases in magnetic photonic crystals (e.g., Nat. Commun. 11, 3119 (2020); Natl. Sci. Rev. 11, nwae121 (2024)) rely on magneto-optical effects, whereas this work achieves quadrupole topology without magnetism. What is the fundamental difference between those two methods? Please explicitly compare them.

Response: We would like to thank the reviewer for the comment. The symmetry constraints on the Wannier band polarizations are different between those two methods. For quadrupole topological phases in magnetic photonic crystals, the Wannier band polarizations are quantized by the combined operation of mirror and time-reversal symmetries M_xT and M_yT , which subsequently leads to the quantized quadrupole moment q_{xy} . While the topological PhCs in our methods host quantized quadrupole moments and composite Wannier bands polarizations under the action of glide symmetries G_x and G_y . The symmetry constraints on Wannier bands and Wannier band polarizations can be deduced by analyzing the transformation properties of the Wilson loop and nested Wilson loop. Consider the Wilson loop along x , the symmetry G_x imposes constraints on the Wilson loop eigenvalues as $e^{i2\pi v_x^j(k_y)} \stackrel{G_x}{=} e^{i2\pi(-v_x^j(k_y)+\frac{1}{2})}$, i.e., $v_x^j(k_y) \stackrel{G_x}{=} -v_x^j(k_y) + 1/2 \pmod{1}$, indicating the Wannier bands come in pairs as $v_x(k_y)$ and $-v_x(k_y)+1/2$. Similarly, the Wilson loop eigenvalues satisfy $e^{i2\pi v_x^j(k_y)} \stackrel{G_y}{=} e^{i2\pi(v_x^j(-k_y)+\frac{1}{2})}$, i.e., $v_x^j(k_y) \stackrel{G_y}{=} v_x^j(-k_y) + \frac{1}{2}$ under the action of G_y . Hence, the Wannier bands occur in $v_x(k_y)$, $-v_x(k_y)+1/2$, $-v_x(-k_y)$, and $v_x(-k_y)+1/2$. Meanwhile, for the constraints from G_x and G_y on the Wannier bands polarizations $p_y^{v_x^j}$, they obey $p_y^{v_x^1}(k_x) \stackrel{G_x}{=} p_y^{v_x^2}(-k_x) + \frac{1}{2}$, $p_y^{v_x^3}(k_x) \stackrel{G_x}{=} p_y^{v_x^4}(-k_x) + \frac{1}{2}$, $p_y^{v_x^1}(k_x) \stackrel{G_y}{=} -p_y^{v_x^3}(k_x) + \frac{1}{2}$, and $p_y^{v_x^3}(k_x) \stackrel{G_y}{=} -p_y^{v_x^4}(k_x) + \frac{1}{2}$, thus provide the quantization of the composite Wannier bands polarizations constructed by Wannier bands “1+3” and “2+4”, namely, nontrivial polarization of $\pm 1/2$ presented in the main text. As demonstrated above, glide symmetry is important in the quantization of Wannier bands polarizations. To further characterize it, we consider a configuration with glide symmetry broken, as shown in Fig. R1. For this configuration, although the Wannier bands are gapped, the polarizations of Wannier sectors “1+3” and “2+4” without quantization due to the strong glide symmetry broken perturbations.

In the revised version, we have added the above discussion of the symmetry constraints on Wannier bands/Wannier band polarizations and the glide symmetry-broken case in the Supplementary Information (Fig. S1), and revised a sentence in the revised main text: For Wannier sectors “1 + 3” and “2 + 4” (orange lines in Fig. 1c), they exhibit nontrivial

polarization of $\pm 1/2$ due to the glide symmetries constraints on Wannier band polarizations (Supplementary Sec. S1).

Figure R1. **a** Wannier band and **b** the nested Wannier bands for the considered glide symmetry broken configuration.

4. The manuscript states that Q -factors increase as the mode volume (V_m) expands, which contradicts the conventional understanding that tighter light confinement (smaller V_m) enhances Q -factors.

Response: We thank the reviewer for the insightful comment. The weaker mode confinement may lead to the light leakage through the domain boundary, namely that the enhanced mode confinement can indeed have larger Q -factors. For the presented topological cavity, although the mode volume increases slightly with the decrease of the defect-dependent parameter Δd_1 , the mode size remains small compared to the finite-size footprint ($\sim 533 \text{ um}^2$), providing sufficient mode confinement to suppress the leakage through the domain boundary. The larger Q -factors for the case with expanded mode volumes in our work can be attributed to the suppression of out-of-plane loss [Nature 425,944 (2003)], which is indicated by the smaller Fourier component in the light cone. To illustrate the influence of out-of-plane loss for Q -factors, Fig. R2 gives the spatial Fourier transformation of the H_z field, where the light cone boundary is indicated by a white circle. It can be observed that the Fourier components inside the light cone boundary for the cases with $\Delta d_1 = 0.25$ [Fig. R2(a)] are more prominent compared with $\Delta d_1 = 0.05$ [Fig. R2(b)], indicating the former hosts more out-of-plane radiation loss, thus resulting in larger Q factors for the cases with $\Delta d_1 = 0.05$ despite the expanded mode volume.

In the revised version, we have added the above discussion in Sec. S5 of Supplementary Information (Fig. S8) to clarify the enhanced Q -factor as the mode volume expands, and the discussion is added on page 7 in the revised main text: The Q -factors are on the order of 10^4 to near 10^6 with the out-of-plane loss significantly suppressed, in which the larger Q -factors for the case with expanded mode volumes can be attributed to the suppression of out-of-plane loss⁴⁵, which is indicated by the smaller Fourier component in the light cone (Supplementary Sec. S5).

Figure R2. Spatial Fourier transformation for the case with **a** $\Delta d_1 = 0.25$ and **b** $\Delta d_1 = 0.05$.

5. The optical pumping setup lacks clarity such as what is the pump spot size and its spatial overlap with the corner state.

Response: For greater clarity regarding the optical pumping setup, we added the optical image in the revised Supplementary Information, showing the pumping laser spot with a diameter of around $2\ \mu\text{m}$ that is close to the size of spatial distribution for the corner state, and it is precisely positioned at the nanocavity by using piezoelectric nanopositioners. The information of pump spot size and Figure R3 is added in the Methods Section and Supplementary Information (Fig. S9), respectively.

Figure R3. Topological nanolasers in a home-made $\mu\text{-PL}$ system. The pumping laser spot is shown in the image, indicating a diameter of around $2\ \mu\text{m}$ and is precisely positioned at the nanocavity by using piezoelectric nanopositioners.

6. In Fig. 3b, the light-in-light-out (L-L) curve exhibits an indistinct threshold kink when pumping power increases. Can the authors explain why?

Response: We thank the reviewer for the comment. In Fig. R4, we plot the $L-L$ curve over a larger range of input power and correct the aspect ratio of the image to exhibit a threshold kink. Moreover, we provide the lasing spectra, $L-L$ curve and linewidth for the representative nanolaser with different magnitudes of defect parameter to further show the lasing behavior with distinct kink in the $L-L$ curves (Fig. R5). In the revised manuscript, we revise Fig. 3a over a larger range of input power, replace Fig. 3b from Fig. R4, and add Fig. R5 in the Supplementary Information (Fig. S16).

Figure R4. $L-L$ curve and linewidth under various input powers

Figure R5. Lasing spectra, L - L curve and linewidth for the proposed nanolasers with defect parameter **a** $\Delta d_1=0.18$, **b** 0.17, **c** 0.16, and **d** 0.15.

7. The temperature-dependent lasing spectra (Fig. 3e) shows no evidence of bulk modes at all. Can the authors explain why?

Response: We thank the reviewer for the comment. For temperature-dependent lasing spectra, we focused on the corner mode in which the pumping spot is spatially located at the corner site rather than the bulk region, thus only the corner state is expected to be excited. To clarify this issue, we revised the sentence on page9: Furthermore, the thermal stability for the proposed defect-evolved quadrupole topological corner state nanolaser is evaluated by placing the devices on a TEC stage. The fabricated device presents stable single mode emission as temperature increases from 30 to 70°C (Supplementary Fig. S11). And according to the comment of Reviewer #3, the discussion of temperature-dependent measurements is moved into the Supplementary Information (Sec. S8) in the revised version.

Reviewer #2

Reviewer #3

The development of quadrupole topological lasers is a timely and important research topic. The idea of defect evolution to realize quadrupole phases could potentially enrich the field of topology-based nanophotonics. However, while the lasing operation in experiment itself appears convincing, I find that the topological origin of the observed lasing remains insufficiently demonstrated. At present, the results could be interpreted as stemming from other conventional localized modes rather than quadrupole topological states.

The suggested structure in the manuscript still raises a specific concerns, such as chiral symmetry, robustness, and quadrupole-based lasing. Related to the comment above, I have some comments for clarification.

Response: We thank the reviewer for the valuable comments and constructive suggestions on our work that highlighted aspects that were not clearly addressed in our initial submission. In the following, we provide specific responses to each of the comments and suggestions, and sincerely hope that the revisions have addressed the questions and concerns.

Comment 1.

The manuscript lacks a clear explanation of where the quadrupole topological mode emerges within the photonic band structure. Specifically, it is essential to identify whether the edge and corner modes lie inside or outside the light cone. Given that modes inside the light cone are susceptible to significant radiation losses, understanding their position is crucial, especially in the context of laser performance. The authors are encouraged to present the band structure overlaid with the light cone and to indicate the frequency and momentum location of the corner modes, preferably in relation to the gap illustrated in Fig. 1d.

Response: We thank the reviewer for the valuable and constructive suggestions. We fully agree that it is essential to identify whether the edge and corner modes lie inside or outside the light cone, especially in the context of laser performance. Figures R6a-b present the spatial Fourier transformation of the H_z field for the corner and edge states, in which the white circle represents the light cone boundary. It can be observed that the Fourier components of the corner and edge states are mainly distributed outside the light cone,

implying the suppressed radiation losses for the designed cavity. Furthermore, Fig. R7 shows the band structure for the representative clockwise evolution case, with the gray and yellow regions representing the light line and bandgap, respectively. The bandgap shown in Fig. R7 hosts the quantized quadrupole moment, and quantized Wannier band polarization for the Wannier sectors (Fig. 1d in the manuscript), providing an opportunity for the corner state hosted in this band gap. i.e., the corner state will be emerging in this gap (red dashed line in Fig. R7). The frequency distribution for different eigenstates under the finite-size system also can be clearly seen in eigenfrequency spectra in Fig. R17a.

Moreover, Fig. 1d in the manuscript shows the k_y -resolved polarizations $p_x^{v_y}(k_y)$ of Wannier bands to describe the feature for the photonic gap. The polarizations of Wannier band are commonly utilized to describe the quadrupole topological phase, the orange line labeled 1+3 in Fig. 1d demonstrated that the Wannier sectors 1+3 (orange line in Fig. 1c) host the quantized Wannier band polarization $1/2$, indicating the quantized quadrupole moment $q_{xy} = 2p_y^{v_x} p_x^{v_y}$ [Science 357, 61 (2017)] and quadrupole topology for the considered gap in the photonic band structure.

In the revised version:

1. We have added the discussion of the spatial Fourier transformation of the H_z field for the corner and edge states in the Supplementary Information (Fig. S7) to show the Fourier components are mainly distributed outside the light cone, and added a sentence in the main text: The Q -factors are on the order of 10^4 to near 10^6 with the out-of-plane loss significantly suppressed, in which the larger Q -factors for the case with expanded mode volumes can be attributed to the suppression of out-of-plane loss⁴⁵, which is indicated by the smaller Fourier component in the light cone (Supplementary Sec. S5).
2. We added the discussion of the eigenfrequency spectra to show the frequency location of different eigenstates in the revised main text.
3. Fig. 1(b) is redesigned to highlight the light cone, and we added a sentence in the caption of Fig. 1(b): The gray and yellow regions represent the light line and bandgap, respectively. The red dashed line indicates the frequency of corner state.
4. To further clarify Fig. 1d, a sentence on page 5 is revised to: To further describe the topological feature for the photonic gap, Fig. 1d shows the k_y -resolved polarizations $p_x^{v_y}(k_y)$ of Wannier bands. For Wannier sectors “1+3” and “2+4” (orange lines in Fig. 1c), they exhibit nontrivial polarization of $\pm 1/2$.

Figure R6. Spatial Fourier transformation for the **a** corner and **b** edge states. [Fig. R2 is the spatial Fourier transformation for the corner state under different Δd_1 that amplified to the vicinity of the light cone and normalized.]

Figure R7. Band structure. The gray and yellow regions represent the light line and bandgap, respectively. The red dashed line indicates the frequency of corner state.

Comment 2.

To verify a topological transition driven by the defect parameter Δd , it would be helpful to track the evolution of the photonic band structure as a function of Δd . The authors need to perform such a scan and to visualize the modal evolution, particularly at high-symmetry points such as the M point. Including mode profiles and phase distributions (e.g., $\arg(E_{\{z\}})$) for both trivial (supercell, $\Delta d=0$) and topological phases would help clarify whether band inversion or gap opening occurs. At present, the distinction between trivial and topological phases is not clearly visible in the data.

Response: We thank the reviewer for the comment. To clarify the topological phase transition of the quadrupole topology in our work. Figure R8a shows the width of the band gap at the Γ point as a function of Δd_1 . The band gap becomes narrower as Δd_1 decreases, whereas the process for topological phase transition without gap closure. The representative field distribution of the M point for the 3rd and 4th bands below the gap and the 5th and 6th bands

above the gap is shown in Figures R8b-c and d-e, respectively. However, different from most topological phase transitions, the band reversion is not a requirement to distinguish a quadrupole topological phase, the topological distinction and feature for the quadrupole topological phase could be commonly manifested by the Wannier band rather than the bulk band [e.g., Phys. Rev. Res 3, 013239 (2021), Nat. Commun. 11, 65 (2020), Phys. Rev. B 102, 035105 (2020), Laser Photonics Rev. 14, 2000010 (2020)]. To shed light on this behavior for the quadrupole topology, Fig. R9 shows the evolution of the polarization of Wannier bands with the defect parameter Δd_1 . Here, the topological phase with negative Δd_1 is labeled as phase I, and the topological phase with positive Δd_1 labeled as phase II. Both of which have nontrivial quadrupole moments but are topologically distinguished by mutually flipped signs of Wannier band polarization. In the revised version, we included the results of the width of the band gap at the Γ point as a function of Δd_1 in the Supplementary Information (Sec. S4) and replaced Fig. 1e from Figure R9. The sentences on page 5 in the main text are revised to: Meanwhile, the nontrivial quadrupole topological phases also occur in counterclockwise configuration, but they are topologically distinguished from clockwise cases by the mutually flipped signs of the Wannier band polarization shown in Fig. 1e and indicate the phase transition at $\Delta d_1=0$. These results indicate that the evolution of defects in different directions in our scheme hosts distinct branches of quadrupole topology.

Figure R8. **a** The width of the band gap at the Γ point as a function of Δd_1 . **b-c** The H_z field distribution for **b** $\Delta d_1=0.25$ and **c** $\Delta d_1=-0.25$ at the M point below the gap. **d-e** The H_z field distribution for **d** $\Delta d_1=0.25$ and **e** $\Delta d_1=-0.25$ at the M point above the gap.

Figure R9. The evolution of the Wannier band polarization with the defect parameter Δd_1 .

Comment 3.

It is not clear whether the simulations shown in Fig. 1b, Fig. 2d, and Fig. 2e were performed using a 2D approximation or full 3D finite-element calculations. Given the slab nature of the experimental system, a thickness is critical for accurate modeling of mode confinement, radiation losses, and Q factor.

Response: The simulations shown in Fig. 1b, Fig. 2d, and Fig. 2e were performed using 3D finite-element calculations. To clarify this issue, we have now revised the following sentence in the Methods Section: In numerical studies, we conduct 3D simulations to obtain the photonic bulk and projected band structure, eigenfrequency spectra, Q -factor, V_m , and field profile by the finite element method. In the 3D simulations, the refractive index of the gain slab is set to $n = 3.25$. To compute the Wannier band and nested Wannier bands, 2D simulations are performed with an effective refractive index of $n = 2.6$. The mode volume V_m of the topological nanocavity is calculated by $V_m = \frac{\int \varepsilon(\mathbf{r})|E(\mathbf{r})|^2 dV}{\max(\varepsilon(\mathbf{r})|E(\mathbf{r})|^2)}$, where $\varepsilon(\mathbf{r})$ and $E(\mathbf{r})$ denote the spatially dependent dielectric constant and electric field intensity, respectively.

Comment 4.

While the authors mention the robustness of the corner state against disorder (see Supplementary Information), it remains unclear whether the structure indeed exhibits genuine topological protection. In many topological systems, such protection arises from underlying chiral or crystalline symmetries. The manuscript should address whether the corner states are pinned to midgap via symmetry constraints or are simply localized modes that are weakly perturbed by disorder. If topological protection is claimed, a clearer explanation of the protecting symmetry and the nature of the topological invariant is needed.

Response: We thank the reviewer for the comment, and we fully agree that in many topological systems, such protection arises from underlying chiral or crystalline symmetries.

For most higher-order topological insulators, the chiral symmetry leads to the corner state pinned to zero energy, and the bulk polarization or quadrupole moment is generally quantized by crystalline symmetries. Nevertheless, the chiral symmetry is always broken in photonic crystals [Nanophotonics 9, 547 (2020)], and crystalline symmetries generally do not inherently constrain the energy of boundary states. For our system, the Wannier bands polarization is quantized by the glide symmetries, which subsequently leads to the quantized quadrupole moment. In the following, we consider a glide symmetry broken configuration as shown in Fig. R10 (also discussion in Fig. R1). the polarizations of Wannier sectors “1+3” and “2+4” without quantization due to the strong glide symmetry broken perturbations.

Furthermore, the robustness against defects is one of the main characteristics of topological photonic systems. The procedure of the previous mention of the robustness of the corner state we followed is similar to that shown in previous studies [e.g., Light:Sci. Appl. 9, 109 (2020), Opt. Express 29, 30735 (2021), Phys. Rev. Lett. 134, 116607 (2025)], particularly for the works of semiconductor material-based systems. And the robustness against defects is further confirmed experimentally. Figure R11d-f shows SEM images of three configurations of nanolasers. Despite these introduced defects, lasing from these topological corner-state modes is sustained, satisfies the expectations of the previous simulation of robustness. These results have also been included in the Supplementary Information (Fig. S3), and we also have added the discussion of the Wannier band polarizations for the glide symmetry-broken cases in the Supplementary Information (Fig. S1).

Figure R10. Nested Wannier bands for the considered glide symmetry broken configuration.

Figure R11. Several defect types are taken into account, including bulk defects **a**, bends **b**, and exchange **c**. **d-f** SEM image of fabricated samples. **g-i** L - L curve and lasing spectra for **g** bulk defects, **h** bends, and **i** exchange.

Comment 5.

The Q-factor plotted in Fig. 2e shows an inverse relationship with Δd_1 , with the highest Q obtained as Δd approaches zero(=the trivial lattice limit). This observation raises the question whether the trivial design may, in fact, be more favorable for laser operation. If no protection mechanisms exist to ensure robustness or midgap pinning, and if the topological design results in lower Q, the motivation for using a quadrupole lattice in a laser becomes unclear. The authors should comment on this trade-off and explain whether the quadrupole configuration offers any definitive advantage for conceptual novelty.

Response: We thank the reviewer for the comment. As Δd_1 decreases, the band gap becomes

narrower [Fig. R8 (a)], although the Q-factor exhibits an inverse relationship with Δd_1 . Excessive narrow band gaps tend to be nonconductive to the implementation of in-gap state lasing. The experimentally lasing behavior upon the defect evolution shown in Fig. 4 of the main text, it can be observed that the lasing peak emerges until the magnitude of Δd_1 evolves to 0.18. Moreover, $\Delta d_1=0$ corresponds to the phase transition point, and the corner state is generated in the single corner cavity, which is composed of two topologically distinguished PhCs ($\Delta d_1 > 0$ and $\Delta d_1 < 0$). Here, $\Delta d_1 > 0$ and $\Delta d_1 < 0$ correspond to two topologically distinct phases with nontrivial quadrupole topology, i.e., both of which have nontrivial quadrupole moments but are topologically distinguished by mutually flipped signs of Wannier band polarization (Fig. R9). On the other hand, the configuration with $\Delta d_1=0$ can be seen as the square lattice with C_{4v} symmetry, which may host Brillouin zone folding bound states in the continuum [e.g., Nat Commun 14, 2811 (2023), PhysRevB 109, 115109 (2024)] that are distinct from the focused corner state in the quadrupole topological phase. The study of Brillouin zone folding bound states in the continuum with defect evolution lies outside the current scope and primary objective of the paper, which we plan to include in our future work.

Comment 6.

The manuscript lacks simulations or experimental discussion of edge states. In higher-order topological insulators, the edge states is often essential in confirming the hierarchy of bulk-edge-corner correspondence. The authors are encouraged to simulate or measure the edge configurations between topologically distinct domains. Simulations at 1D interfaces (edges) would help resolve this issue.

Response: We thank the reviewers for pointing out the lack of edge states in the simulation or experimental discussion. In the revised manuscript, we have provided information on the projected band structure and the field distribution of the simulation edge state for ribbon-like supercells in the revised main text (Fig. 2d), and the SEM, L - L curve, and lasing spectra of the fabricated devices with a flat interface have been included in the Supplementary Information (Fig. S2). Correspondingly, the discussion is added on page 6: Before numerically studying the corner state, we first verified the edge states at the topologically distinct domains in ribbon-like supercells. Fig. 2d shows the projected band structure that reveals the 1D edge states (purple lines) inside the bulk gap. The field distribution for the typical lower frequency edge state is plotted in the top panel of Fig. 2d. The electromagnetic field is strongly localized at the topologically distinct domain. Topological 1D edge state lasing emission from the flat interface can be realized and has been experimentally verified, as shown in Supplementary Sec. S2.

Figure R12. **a** Projected band structure and the field distribution of the simulation edge state for ribbon-like supercells. **b** SEM image of fabricated sample with flat interface. **c** L - L curve and lasing spectra.

Comment 7.

To better understand the modal characteristics, I suggest the authors simulate the far-field patterns and analysis their polarization dependence for the simulated lasing mode. Quadrupole states might be expected to exhibit unique field symmetries and polarization features. These could help distinguish them from other possible defect-localized modes. Providing polarization-resolved near-field or far-field measurements would also be valuable.

Response: We thank the reviewer for the comment. The corner state here is generated in a single corner cavity under the quadrupole topological phase, which is different from the quadrupole bulk mode [e.g., Phys. Rev. A 100, 063803 (2019), Nat. Nanotechnol. 15, 67 (2020)] or the quadrupole mode by coupled corner states [e.g., Nat Commun 11, 5758 (2020)]. To better understand the modal characteristics, we experimentally obtain the far field of corner and edge state lasing shown in Fig. R13, and compared with the simulation far field shown in Fig. R14. One can observe that the experimental far-field pattern of corner state lasing exhibits unique field distribution distinct from the edge state lasing, although the proposed topological nanolasers wavelength-scale mode volume ($\sim 0.62(\lambda/n)^3$) that leads to the highly divergent far-field emission. Meanwhile, the experimental far field profiles for both corner state and edge state agree well with the simulated results, and help to distinguish the corner emission. These results are also provided in the Supplementary Information (Sec.10) for the revised version.

Figure R13. Far-field patterns for **a** corner lasing, **b** edge lasing from the horizontal interface, and **c** edge lasing from the vertical interface.

Figure R14. Simulated far-field patterns for **a** corner state, **b** edge state from the horizontal interface, and **c** edge state from the vertical interface. Weak disorder is introduced in the vertical interface to reduce the C_4 symmetry.

Comment 8.

Related to the previous point, it would be helpful to show polarization-resolved near-field images of the lasing emission and to compare these with the simulated quadrupole mode profiles. I hope that such measurements would offer further insight into whether the lasing mode indeed arises from the expected quadrupole corner state. Additionally, further verification is needed to ensure that the observed lasing mode is not a fundamental mode (1st band) arising from non-topological band folding or structural asymmetry

Response: We thank the reviewer for the comment. There is no obvious difference for the x and y components of electric fields for the corner state in our structure [Fig. R15], and the wavelength for the fundamental mode of the 1st band is far from 1550 nm. Moreover, Fig. R16 exhibits the polarization-angle-dependent output intensity that shows linearly polarized emission from the lasing mode. In order to excite the single-mode lasing action attributed to the corner state, we selectively pump the corner site and use an InGaAs camera to image the optical profiles as shown in Fig. 3c, in which the emission spot is well localized at the corner region, confirming the lasing action from the corner state. Figure R17a shows eigenfrequency spectra of the investigated cavity, and Fig. R17b shows the lasing spectra with selectively excited corner, edge, and bulk states by choosing distinct optical excitation spots that are located at the corner, edge, and bulk, respectively. The laser behavior, including expected topological mode selection and spectrum trends, follows the expected behavior for the corner, edge and bulk states, further confirming that the previous emission comes from the corner state. Moreover, the far-field characterization in Fig. R13 also indicates the characteristics of the corner state.

To avoid the confusion with quadrupole bulk mode and quadrupole corner state that means quadrupole mode by coupled corner states, we revised the term “quadrupole topological corner state” to “corner state in quadrupole topological phases” and “quadrupole topological corner state nanolaser/nanocavity” to “quadrupole higher-order topological PhC

nanolaser/nanocavity”. The results of eigenfrequency spectra of the investigated cavity, near-field optical profiles of edge and bulk lasing, and measured lasing spectra with the optical excitation spot located at the corner, edge, and bulk regions are added to the revised main text as Fig. 2e, Fig. 3d-e, and Fig. 3f. The polarization-angle-dependent output intensity is included in the Supplementary Information (Sec. S7) for the revised version. And we added the following sentences on page 9 in the main text: To excite the lasing from edge and bulk states, the optical excitation spot is selectively located at the corresponding edge and bulk region, respectively. Figs. 3d and 3e depict the near-field optical profiles of edge and bulk lasing below (top panel) and above (bottom panel) lasing thresholds, and Fig. 3f shows the lasing spectra for different states (the collected L - L curves of edge and bulk states are shown in Supplementary Sec. S9). The lasing behavior, including the anticipated topological mode selection and spectral trends, is consistent with the expected characteristics of the corner, edge, and bulk states. Moreover, the far-field pattern of corner state lasing exhibits a unique distribution, distinct from that of edge state lasing (Supplementary Fig. S14), thereby confirming the lasing operation from the designed corner state.

Figure R15. **a** x component and **b** y component of electric fields for the corner state.

Figure R16. Polarization-angle-dependent output intensity for the topological corner-state nanolasers.

Figure R17. **a** Eigenfrequency spectra of the investigated cavity. **b** Measured lasing spectra with the optical excitation spot located at the corner, edge, and bulk, respectively. The intensity from the bulk is magnified 10 times for clarity. **c** The optical excitation spot for the edge and bulk regions.

Comment 9.

It is unclear how temperature-dependent measurements (Fig. 3f, 3g) relate to the topological aspects of the design. Is there any reason to expect a topological invariant or corner-state localization to degrade (or persist) with temperature? To my knowledge, it is confusing why the authors showed the thermal dependence for quadrupole lasing operations.

Response: We thank the reviewer for the comment. The purpose of the temperature-dependent measurements is to study the thermal stability of the lasing behavior of the fabricated device. To avoid confusion, the discussion of temperature-dependent measurements is moved into the Supplementary Information in the revised version.

Comment 10.

For further insight into the lasing behavior, the authors might consider varying the duty cycle of the pump pulses by increasing the pulse width at fixed repetition rate. This would help assess thermal effects, and potentially the CW response of the lasing mode. While not essential, such measurements could enhance the depth of characterization.

Response: According to the suggestion, we have provided the lasing behavior of the proposed device by increasing the pulse width at a fixed repetition rate of 200 kHz. The normalized lasing spectra under various pulse widths are shown in Fig. R18a, exhibiting stable single mode lasing as pulse width increases. Figure R18b shows the collected L - L curves under various pulse width. The significantly increased laser threshold under larger pulse width can be attributed to the reduced gain and increased non-radiative recombination rates. For further

insight into the lasing behavior, we also included the time-resolved photo-luminescence (TRPL) measurements to investigate carrier dynamics. Figure R19a displays normalized TRPL spectra for a representative nanolaser, exhibiting a clear transition from spontaneous emission to the coherent stimulated emission with a shorter stimulated lifetime ($\tau_{\text{lasing}}=0.26$ ns). The above results have included in the Supplementary Information (Fig. S11 and Fig. S12).

Figure R18. **a** Pulse width-dependent normalized lasing spectra. **b** Pulselength-dependent normalized L-L curves.

Figure R19. **a** Normalized TRPL spectra of the spontaneous emission (orange dots) and stimulated emission (pink dots). **b** L-L curve and lasing spectra for the corresponding representative nanolaser.

While the reported results are promising in terms of device functionality and experimental feasibility for lasing, the connection between lasing and quadrupole topology is not sufficiently substantiated at this stage. I encourage the authors to address the above points (notably, concerning the topological origin of the lasing mode). Solving those clarifications, I will look forward a definitive demonstration for a next round of revision.

Response: We thank the reviewer again for the constructive comments and acknowledge the time and effort the reviewer has spent in assessing our work. We have revised the paper based on the feedback and comments from the reviewers and hope that we could adequately address the primary concerns.

Reviewer #1 (Remarks to the Author):

I appreciate the authors' efforts during the revision. The previous technical concerns have been resolved. The authors have introduced a novel approach for constructing a photonic quadrupole insulator. This method does not rely on tight-binding models such as the 2D SSH model, nor does it require magnetic materials, making it highly promising for photonic applications such as topological lasers as demonstrated in this work. I recommend the manuscript for publication.

Reviewer #3 (Remarks to the Author):

This manuscript introduces an topologically quadrupolar design for topological nanolasers. However, I still have several key concerns:

- (1) The position of the mode relative to the light cone is not clearly resolved.
- (2) The connection between the simulated Wannier-like topology and the lasing mode remains indirect.
- (3) I am not sure that the experimental evidence does not yet clearly support the interpretation of a quadrupole HOTI mode.

I appreciate the authors' efforts for high quality experimental results. However, the origin of quadrupolar topology and their experimental lasing system remains insufficiently demonstrated. Below, I provide detailed comments for addressing my concerns.

Comment 1.

Most central and critical issue is that the claimed quadrupole HOTI mode seems to form a frequencies in above-light cone regime. While Wannier-like photonic system may be allowed in theory or idealized simulations, but in practice, light cone and its position in photonic bands leads to substantial radiation loss, and limit the Q factor and making laser operation extremely challenging.

Most successful demonstrations of photonic topological lasers operate near the fundamental band and their topology because it is clearly positioned below to the light cone. In contrast, the current design places the quadrupole mode well above the light cone at Gamma and X. This reason makes it difficult whether the observed lasing is truly from a topological mode, or a conventional cavity mode. The authors should clearly identify the wavevector and frequency of the quadrupole mode relative to the light cone in the band diagram

Comment 2.

If the mode lies inside the light cone, it becomes important to clarify which wavevector the red dashed line (corner state) in Fig. 1b corresponds to. Specifically, at which momentum point is the band topology formed?

Comment 3.

Although robustness tests are presented (Fig. S3), the perturbations are too far from the corner. This does not adequately test whether the corner mode is topologically protected. Generally, it would be meaningful to introduce bulk disorder including or near the corner region.

Comment 4.

As the authors already addressed in their response, the existence of chiral symmetry in current design should be mentioned in the main text. While there are topological studies on systems with chiral symmetry, such investigations remain relevant even when the symmetry is not strictly preserved in photonic system. It is important to reflect this point in the manuscript as well. In my view, even a single sentence clarifying this in the main text would be sufficient for readers.

Comment 5.

The authors explain on Wannier band polarizations and nested Wilson loops to argue for quadrupole topology. I agree to this theoretical approaches to design a quadrupole in photonics and it already introduced in several times in other reports. However, in current version there still lack clear real-space verification that the experimental lasing mode corresponds to the predicted quadrupole corner state.

Comment 6.

The authors are encouraged to provide unsaturated real-space images of the lasing mode above threshold. To confirm that the observed lasing mode indeed corresponds to a Wannier-based quadrupole topological state, it is essential to analyze the spatial symmetry and vector-field distribution of the mode.

Currently, the mode profile images (Fig. 3c–e) do not provide sufficient information to assess the underlying field profiles, due to the saturation. It is not possible to analyze the in-plane E-field components, which may introduce evidence for identifying the mode's symmetry and matching to theoretical quadrupole corner states model.

Comment 7.

The reported threshold ($\sim 0.5 \mu\text{W}$) appears extremely low and raises concerns about how it was measured. Is the value based on average or peak pump power? This must be clarified in both the text and figure captions. If using average power, convert to peak power for comparison with other nanolaser works. On general understanding, a peak pump power of $0.5 \mu\text{W}$ is extremely low, and it is unlikely that suggested III–V semiconductors would respond at such levels.

Comment 8.

I did not specifically intend to highlight the M point alone, but rather to suggest that the authors identify the conditions under which topologically quadrupole mode inversion occurs, and the corresponding wavevector where it takes place. I mentioned the M point simply as an example, because a mode inversion occurring within the light cone is more likely to result in a physically realizable lasing system.

The current analysis still lacks sufficient mode profile information of the unit cell as a function of wavevector. This makes it difficult to determine under what conditions the E-field distribution can be considered as “Wannier-like”.

Upon reviewing the current data, however, the closest relevant bands seem to lie along the Γ –X direction. I still find it unclear whether the observed lasing mode genuinely originates from a topological Wannier-like system, and whether this system can be convincingly interpreted as a laser based on a quadrupole topological phase.

Reviewer #1

I appreciate the authors' efforts during the revision. The previous technical concerns have been resolved. The authors have introduced a novel approach for constructing a photonic quadrupole insulator. This method does not rely on tight-binding models such as the 2D SSH model, nor does it require magnetic materials, making it highly promising for photonic applications such as topological lasers as demonstrated in this work. I recommend the manuscript for publication.

Response: We sincerely thank the reviewer for the constructive comments on our manuscript throughout the review process and the recommendation of publication.

Reviewer #3

This manuscript introduces an topologically quadrupolar design for topological nanolasers. However, I still have several key concerns:

- (1) The position of the mode relative to the light cone is not clearly resolved.
- (2) The connection between the simulated Wannier-like topology and the lasing mode remains indirect.
- (3) I am not sure that the experimental evidence does not yet clearly support the interpretation of a quadrupole HOTI mode.

I appreciate the authors' efforts for high quality experimental results. However, the origin of quadrupolar topology and their experimental lasing system remains insufficiently demonstrated. Below, I provide detailed comments for addressing my concerns.

Response: We sincerely thank you for taking the time and effort to review our manuscript and for providing insightful comments that have helped us further improve its quality and clarity.

We also appreciate your positive remarks regarding our efforts to obtain high-quality experimental results. Regarding the concern raised about whether the observed lasing originates from the focusing system, we have carried out additional simulations and experiments and provided further clarifications to address this issue. Below, we respond to each comment point by point and have incorporated the corresponding revisions into the revised manuscript.

Comment 1.

Most central and critical issue is that the claimed quadrupole HOTI mode seems to form

a frequencies in above-light cone regime. While Wannier-like photonic system may be allowed in theory or idealized simulations, but in practice, light cone and its position in photonic bands leads to substantial radiation loss, and limit the Q factor and making laser operation extremely challenging.

Most successful demonstrations of photonic topological lasers operate near the fundamental band and their topology because it is clearly positioned below to the light cone. In contrast, the current design places the quadrupole mode well above the light cone at Gamma and X. This reason makes it difficult whether the observed lasing is truly from a topological mode, or a conventional cavity mode. The authors should clearly identify the wavevector and frequency of the quadrupole mode relative to the light cone in the band diagram

Response: We thank the reviewer for this insightful comment. The substantial radiation loss and a limited Q factor may occur if the corner state enters the light cone.

Indeed, the wavevector component of corner state in our work is outside the light cone, which mean the suppressed radiation loss and giving rise to high Q -factors that we have shown in the manuscript. Figure R1 shows the spatial Fourier transforms of the corner state at the corresponding frequency to identify the associated wavevector and frequency relative to the light cone. The wavevector component inside the light cone boundary (labeled by gray region) is negligible compared to the outside. Meanwhile, the wavevector component is concentrated on the extended Brillouin zone rather than within the light cone at X and Γ points of the first Brillouin zone (white square) due to being built on the folded band.

Figure R1. Spatial Fourier transformation for the corner state. The white square represents the first BZ. The gray region represents the light-cone boundary, with the dashed yellow circle representing the cross section of the light-cone boundary at the frequency of the corner state.

The suppressed radiation loss, high Q -factor with small mode volume, and robustness against defects and disorders facilitate the attainment of the lasing emission

from the corner state in our scheme, and naturally tend to stable single-mode operation. To make the explanation of the position of the mode relative to the light cone clearer, we included the above explanation and Fig. R1 in the revised main text and made related modifications in Sec5.

Comment 2.

If the mode lies inside the light cone, it becomes important to clarify which wavevector the red dashed line (corner state) in Fig. 1b corresponds to. Specifically, at which momentum point is the band topology formed?

Response: We thank the reviewer for this relevant comment. As explained in our response above to *Comment 1*, the wavevector component lies outside the light cone, leading to suppressed radiation loss that facilitates the achievement of the lasing emission from the designed corner state. The corresponding revisions have been addressed in our response to the previous comment. Since *Comment 8* below is related to the topological interpretation of the quadrupole topological phase, a detailed explanation of the topology is provided in our response to *Comment 8*.

Comment 3.

Although robustness tests are presented (Fig. S3), the perturbations are too far from the corner. This does not adequately test whether the corner mode is topologically protected. Generally, it would be meaningful to introduce bulk disorder including or near the corner region.

Response: We thank the reviewer for the comment. In the previous revision, we have demonstrated the robustness against defects experimentally by introducing three types of defects, including bulk defects, bends at the boundaries, and exchange of unit cells of the corner region.

Figure R2. Robustness of the corner state by introducing 10% disorder strength into the positions and sizes of air holes in the corner region.

Following the suggestion of the reviewer, we further tested the robustness of the corner state in simulation by introducing disorder in the corner region. As shown in Fig. R2, the 10% disorder strength of air hole positions and sizes in the corner region is introduced, in which the disorder for each hole is independent and performed in 10 test configurations. The corner state undergoes only a slight wavelength shift, showing the robustness against fabrication disorders. In the revised version, we included this result in Supplementary Information.

Comment 4.

As the authors already addressed in their response, the existence of chiral symmetry in current design should be mentioned in the main text. While there are topological studies on systems with chiral symmetry, such investigations remain relevant even when the symmetry is not strictly preserved in photonic system. It is important to reflect this point in the manuscript as well. In my view, even a single sentence clarifying this in the main text would be sufficient for readers.

Response: We thank the reviewer for the constructive comment. In response, we have added a clarifying statement addressing this point in the revised main text on page 7.

Comment 5.

The authors explain on Wannier band polarizations and nested Wilson loops to argue for quadrupole topology. I agree to this theoretical approaches to design a quadrupole in photonics and it already introduced in several times in other reports. However, in current version there still lack clear real-space verification that the experimental lasing mode corresponds to the predicted quadrupole corner state.

Response: We thank the reviewer for raising this point and are glad to clarify. With reference to the *Comments* 5 and 6, we understand that the reviewer expect the lasing mode originating from the single corner state in the quadrupole topological phase to exhibit field features like quadrupolar distribution.

In fact, real-space quadrupolar field distribution is neither a defining feature nor a consequence of the quadrupole topological phase. The so-called real-space quadrupolar distribution phenomenon for corner state is the result from the coupling between corner states in the small-period case [Nat. Commun. 11, 5758 (2020)], which is independent of the quadrupole topological phase. The most studies based on various artificial band-gap systems have well-demonstrated this point [e.g., Nature. Phys 14, 925(2018), Nat. Photonics 13, 692(2019), Nat. Commun 13, 6597 (2022)], in which the real-space quadrupolar distribution is not observed in their mode profile based on the quadrupole topological phase. The direct result of the quadrupole topological phase in real space is

the robust corner states with wavelength-scale mode volume and high Q factor. As explained in the response for Comment 1, the single corner state with a high Q factor and robustness against defects and disorders enables us to obtain the preferential lasing emission with stable single mode operation from the predicted corner state when we pump the corner region. In our experiment, the stable single mode lasing with stronger peak intensities pumped from the corner region and excitation wavelength trends under distinct mode selection are observed (Fig. 3), matching the predicted spectroscopic characteristics of the proposed scheme. In the previous version, the far-field pattern is also discussed to distinguish the corner and edge state lasing (Fig. S14). Also, the evolution trend of lasing wavelength in different devices under the designed manipulation scheme (Fig. 4), as well as stable single mode lasing with different perturbation configurations (Fig. S3), both further underpin that the lasing mode originated from the predicted corner state.

Additional characterization of the lasing behavior is provided in our response to *Comment 6*, where we provide the unsaturated mode profile and position-dependent PL intensities to clarify this point (Fig. R3).

Comment 6.

The authors are encouraged to provide unsaturated real-space images of the lasing mode above threshold. To confirm that the observed lasing mode indeed corresponds to a Wannier-based quadrupole topological state, it is essential to analyze the spatial symmetry and vector-field distribution of the mode.

Currently, the mode profile images (Fig. 3c–e) do not provide sufficient information to assess the underlying field profiles, due to the saturation. It is not possible to analyze the in-plane E-field components, which may introduce evidence for identifying the mode's symmetry and matching to theoretical quadrupole corner states model.

Response: We thank the reviewer for this relevant comment. As we detailed in the response to *Comment 5*, the real-space field feature like quadrupolar distribution is irrelevant to the corner states from the quadrupole topological phase. This is also reflected in the mode profile shown in Fig. R3a for our work, which not exhibits the real-space quadrupolar distribution. Moreover, Fig. R3b shows five representative fabricated devices for horizontal position-dependent normalized intensities, with the inset shows the simulated electric field distribution. One can see the lasing mode is localized in the corner region and shows a slower decay along the edge region direction compared to the opposite direction (the topologically distinct region), which is in good agreement with the simulated result.

To more intuitively illustrate that the lasing mode corresponds to the predicted corner

state, in the revised manuscript, we added the related discussion and included Fig. R3 (replaced Fig. 3c–e) in the main text. The term of the "quadrupole corner state" is deleted to make things clear to a wider range of readers and avoid the misunderstanding that the corner state in quadrupole topological phase is related to the real-space quadrupolar field distribution. Moreover, to focus more on the core object of this study, the near fields of edge and bulk state lasing of the previous version were moved to SI.

Figure R3. **a** mode profile of corner state lasing. **b** position-dependent normalized PL intensities, the inset shows the simulated electric field distribution in horizontal direction.

Comment 7.

The reported threshold ($\sim 0.5 \mu\text{W}$) appears extremely low and raises concerns about how it was measured. Is the value based on average or peak pump power? This must be clarified in both the text and figure captions. If using average power, convert to peak power for comparison with other nanolaser works. On general understanding, a peak pump power of $0.5 \mu\text{W}$ is extremely low, and it is unlikely that suggested III–V semiconductors would respond at such levels.

Response: We thank the reviewer for this importance comment. The reported threshold is based on the average pump power of a 632 nm pulsed laser with repetition rate 200 kHz and duty cycle=0.5%, and the corresponding peak pump power is $100 \mu\text{W}$ (power density $3.18 \text{ kW}/\text{cm}^2$). We apologize for the lacking statement of average or peak pump power. The associate statement and the threshold of peak pump power and power density are clarified in the revised manuscript.

Comment 8.

I did not specifically intend to highlight the M point alone, but rather to suggest that the authors identify the conditions under which topologically quadrupole mode inversion occurs, and the corresponding wavevector where it takes place. I mentioned the M point

simply as an example, because a mode inversion occurring within the light cone is more likely to result in a physically realizable lasing system.

The current analysis still lacks sufficient mode profile information of the unit cell as a function of wavevector. This makes it difficult to determine under what conditions the E-field distribution can be considered as “Wannier-like”.

Upon reviewing the current data, however, the closest relevant bands seem to lie along the Γ -X direction. I still find it unclear whether the observed lasing mode genuinely originates from a topological Wannier-like system, and whether this system can be convincingly interpreted as a laser based on a quadrupole topological phase.

Response: We thank the reviewer for the comment. Different topological phases have diverse topological descriptions. For quadrupole topology, the hallmarks are the vanishing bulk dipole moment and quantized bulk quadrupole moment. Using the framework of symmetry-indicator invariant [Phys. Rev. B 99, 24515 (2019)], i.e., comparing the symmetry representations of the mode profile for occupied bands at high symmetric points, one can capture the feature of vanishing bulk dipole moment. In the previous version, this feature is characterized by the Wannier band, which is a common way for describing it. Under the framework of symmetry-indicator invariant with C_4 symmetry, bulk dipole moment is expressed as $\mathbf{P}^{(4)} = \frac{1}{2}[X_1^{(2)}](\mathbf{a}_1 + \mathbf{a}_2)$, in which C_2 invariants $[X_1^{(2)}] = \#X_1^{(2)} - \Gamma_1^{(2)}$ described the difference in number of occupied bands with symmetry eigenvalue for X and Γ points. The C_2 eigenvalue ± 1 (equivalent to the even /odd parities) at the high symmetric points is labeled in Fig. R4, and since the zero-frequency optical mode is generally even-parity, $\mathbf{P}^{(4)} = (0, 0)$, i.e., vanishing bulk dipole moment, which is consistent with the results of the analysis using the Wannier band and also applicable for both clockwise and counterclockwise evolution cases.

It should be emphasized that the symmetry representation of the mode profile alone is not sufficient for the description of quadrupole topology. Distinguishing of quadrupole topology goes beyond the framework of the symmetry-indicator invariant, which needs to be diagnosed by nested Wannier bands (as we done in the manuscript). In the revised version, we further stress the feature of quadrupole topology and add the above discussion of symmetry representations in the main text. Also, Fig. R4 is included in the main text.

To address the concern of whether this system can be convincingly interpreted as a laser based on quadrupole topological phases, we have provided the information and clarifications of the corner state relative to the light cone (*Comment 1*), more clarifications on the quadrupole topology and lasing (*Comment 5*), as well as the extra

optical characterization (*Comment 6*). We would like to reiterate our sincere gratitude for your constructive comments, and we hope this revision will be more clearly presented to address the concern you raised.

Figure R4. Band structure. The C_2 eigenvalue at Γ and X are labeled as “ \pm ”.